# Transition from predictable to variable motor cortex and striatal ensemble patterning during behavioral exploration

Sravani Kondapavulur [1,2,3,4], Stefan M. Lemke[3,4,5], David Darevsky[1,2,3,4], Ling Guo[3,4,5], Preeya Khanna[3,4] & Karunesh Ganguly [3,4✉]

Animals can capitalize on invariance in the environment by learning and automating highly consistent actions; however, they must also remain flexible and adapt to environmental changes. It remains unclear how primary motor cortex (M1) can drive precise movements, yet also support behavioral exploration when faced with consistent errors. Using a reach-to-grasp task in rats, along with simultaneous electrophysiological monitoring in M1 and dorsolateral striatum (DLS), we find that behavioral exploration to overcome consistent task errors is closely associated with tandem increases in M1 and DLS neural variability; subsequently, consistent ensemble patterning returns with convergence to a new successful strategy. We also show that compared to reliably patterned intracranial microstimulation in M1, variable stimulation patterns result in significantly greater movement variability. Our results thus indicate that motor and striatal areas can flexibly transition between two modes, reliable neural pattern generation for automatic and precise movements versus variable neural patterning for behavioral exploration.

[1] Bioengineering Graduate Program, University of California San Francisco, San Francisco, CA, USA. [2] Medical Scientist Training Program, University of California San Francisco, San Francisco, CA, USA. [3] Neurology and Rehabilitation Service, San Francisco Veterans Affairs Medical Center, San Francisco, CA, USA. [4] Department of Neurology, University of California San Francisco, San Francisco, CA, USA. [5] Neuroscience Graduate Program, University of California San Francisco, San Francisco, CA, USA. ✉email: karunesh.ganguly@ucsf.edu

Motor skills are typically characterized as actions produced in a fast, accurate and consistent manner;[1,2] however, ideally, a skill should also remain flexible in order to readily adjust to changes in task parameters or the environment[3]. How the motor network can flexibly support both the production of a consistent motor skill as well as behavioral exploration to enable error correction is poorly understood. This knowledge gap is particularly the case for perturbations that result in large errors that cannot be corrected through rapid short-term behavioral adaptations[4–8]. In general, primary motor cortex (M1) has been characterized as an engine for precise movement control through the generation of highly stable ensemble spiking dynamics in relation to behavior—e.g., reliable sequential firing[6,9–15]. The dorsolateral striatum (DLS), which receives input from M1 and other cortical regions, is also known to play an essential role in the learning of consistent skilled control through coordination of activity with M1[16–20]; this also results in automatization—task performance with little conscious effort[21–23]. For a consolidated automatic skill[24], it remains unclear how M1 and DLS activity patterns might respond to consistent motor errors. Here, we are particularly interested in how M1 and DLS might support modification of skill components (e.g., the specific endpoint goal) while still largely maintaining the original action (i.e., reach-to-grasp skill in rats[25–28]).

What changes in M1 and DLS activity patterns could occur when a previously successful and automatic action can no longer achieve a goal? One possibility is that skill learning establishes a generalizable neural manifold across M1 and DLS, such that the learned automatic action is modified to achieve a new goal by largely leveraging existing task activity patterns[4]. In this case, we would expect M1 and DLS units to remain strongly modulated in a task-specific manner (e.g., well-modulated by reach onset) and to maintain cross-area coupling. A second possibility is that M1 and DLS could, in tandem, revert to a variable neural state, to enable behavioral exploration, subsequently reestablishing a cross-area manifold for the modified skilled action. In contrast with the first case, we would then expect M1 and DLS units to temporarily lose consistent task-related modulation even while the action is grossly preserved, signaling a process of enhanced neural variability during behavioral exploration and "relearning" of the skilled action.

Here, we studied how behavioral flexibility emerges when an automatic skill results in consistent errors in a reach-to-grasp task in rats. Animals first achieved reliable skilled performance for a single pellet location, during which we observed, as expected, highly reliable M1-DLS activity patterns. Subsequently, we introduced a new pellet location, and training continued over days. Notably, animals learned the modified task across days, rather than within a session—thus, establishing that behavioral exploration from an automatic state was not due to rapid adaptation. Strikingly, establishment of behavioral exploration was closely associated with a transition from highly predictable M1 and DLS ensemble activations to a variable state in both regions, marked by a significant decrease in task-related spiking modulation and greater across-trial pattern variability. We also noted a drop in M1 and DLS cross-area subspace (CS) modulation during reaching, measured using the dimensionality reduction technique canonical correlation analysis (CCA), which defines axes of maximal correlation between neural populations. With successful relearning, M1 and DLS, in tandem, regained task-aligned spiking with corresponding increased CS modulation. Finally, we examined how neural variability in M1 may drive behavioral exploration. We found that compared to reliably patterned intracranial microstimulation (ICMS) in M1, variable ICMS patterns resulted in significantly greater endpoint variability. Together, our results indicate that tandem increases in M1-DLS neural variability are closely linked to behavioral exploration. More broadly, we demonstrate that M1, in conjunction with DLS, can flexibly transition between two distinct modes, reliable pattern generation for precise and automatic skilled control and variable patterning for focused behavioral exploration within an established skill when faced with consistent errors.

## Results

**Modification of an automatic skill is a multiday process.** We simultaneously recorded single-unit activity and local field potentials (LFPs) in M1 and DLS as rats ($n = 6$) learned a reach-to-grasp skill to a single pellet location (Location A, Fig. 1a, b). Training was conducted using an automated behavioral apparatus; presentation of an auditory cue and door opening served to instruct the animal to initiate reaches[29]. Either a single or two cameras were used to monitor kinematics (see Methods). Each animal was initially trained for at least 1250 trials (range of trials = 1250 to 3020, Supplementary Fig. 1; ~100–150 trials per day, range = 9–16 days). Neural activity in M1 and DLS was recorded after reaching to location A was at >50% successful pellet retrieval and reach speed was stable ("baseline"). We then switched the pellet location (Location B); notably, we varied the physical pellet locations of B across animals in order to ensure that locations A and B were not systematically at a central versus lateral position ($n = 2$ central A with switch to lateral B and then $n = 4$ lateral A with switch to central B). After the reach location was switched to Location B, we continued to train animals for 50–150 trials per day for a period of 3–5 days (Fig. 1b–d). Importantly, task accuracy on the single pellet reach to grasp task is highly dependent on the reach component. Thus, flexible switching to Location B first requires successful reaching to the new location (i.e., prior to consideration of grasping); our subsequent quantification of behavioral variability was based on endpoint reach location (Fig. 1c). We observed individual differences in "rates of decay" in perseverance to location A when the pellet location was moved (Fig. 1d); this likely indicates how automatic an animal is after the initial training period. We also fit a function to quantify the rates of decay (Supplementary Fig. 1b). Interestingly, estimates of decay rates were closely correlated with the total number of trials that the animals had been continuously trained to Location A prior to the switch of pellet location (Supplementary Fig. 1b, c, $R^2 = 0.729$, $p = 0.03$).

At a macroscopic level, we observed that reliable switching of reaches to location B was a multiday process (Fig. 1c, d) that involved a transition from initiating trials to A to initiating trials to B, with little evidence of within-trial switching and or outcome-based switching from trial-to-trial (Fig. 1e–i, Supplementary Movie 1). To assess for within-trial switching, we analyzed the characteristics of secondary reaches (i.e., after the first reach attempt) within a trial (Fig. 1e). We characterized secondary reaches as Location A, Location B, and "short reaches" (S, reaches that crossed the door but did not reach the pellet, Fig. 1f). In the first session post-switch, rats perseverated in reaching to Location A (Fig. 1g, ANOVA, Tukey-Kramer post hoc test, A > S, $p = 4.3e{-}4$; A > B, $p = 0.023$). This perseveration did not happen on the few trials when the first reach was to Location B (ANOVA, Tukey-Kramer post hoc test, B > A, $p = 2.2e{-}3$; B > S, $p = 1.1e{-}3$). Thus, when animals initiated a reach to A or B, they continued to reach to the same location within a trial. Finally, we probed whether early post-switch error-updating happened from trial to trial based on accurate retrieval of a pellet (Fig. 1h). After reaching inaccurately to Location A, perseveration to A still occurred on the next trial (Fig. 1i, ANOVA, Tukey-Kramer post hoc test, A > S, $p = 1.5e{-}3$; A > B, $p = 0.01$), indicating that, in the immediate post-switch period,

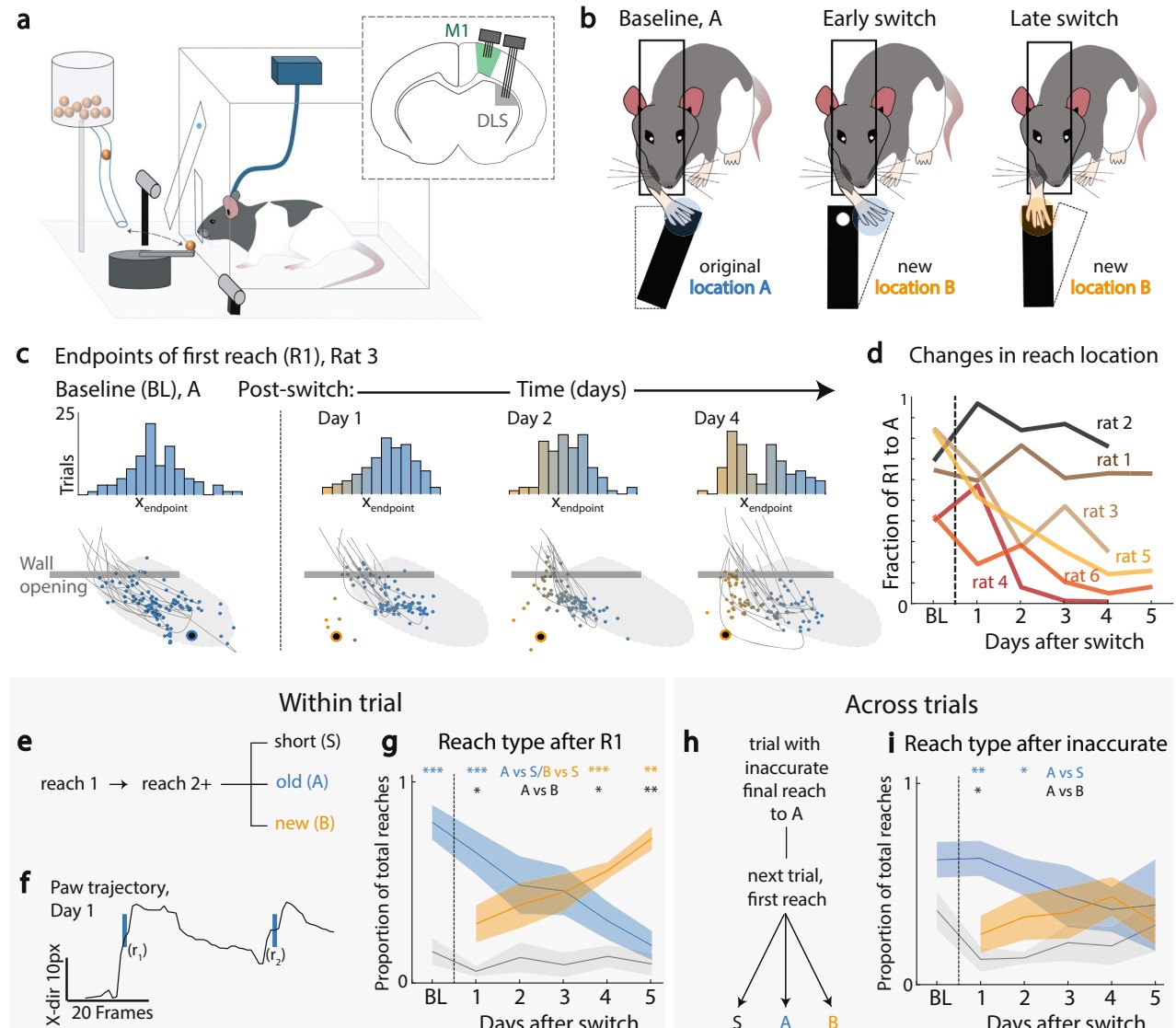

**Fig. 1 Multiday relearning of an automatic skill. a** Automated reach-to-grasp setup. After the door opens (signaled by a tone), the rat reaches through a slit to retrieve the pellet. M1 (green) and DLS (gray) locations in inset. **b** Relearning paradigm. Rats are overtrained to reach to location A (left). Then pellet moved to location B with continued training (middle, right). **c** Endpoint of first reaches. Top: example animal, histogram of endpoint x-position, across trials. Bottom: example reach trajectories and endpoint locations relative to pellet location (large black circle) for same sessions as (top). **d** Decay in reaches to A. Percentage of trials in a session with first reach (R1) to location A, as compared to low-amplitude (short, S) reaches and/or reaches to location B. **e** For those trials with multiple reaches, within-trial updating for reaches after the first were classified into short (S), old (A), or new (B) reaches. **f** Example x-trajectory of paw during a trial, with reach onset marked; r1 and r2 are the first and second reach onsets, respectively. **g** Reach type after first reach, for trials with multiple reaches. For all reaches after the first reach in a trial, proportion of A, B, or S reaches. Data are presented as mean values ± SEM. **h** Following an inaccurate trial, we classified the first reach type of the subsequent trial**. i** Reach type after inaccurate reach to A on previous trial. Data are presented as mean values ± SEM. *<0.05; **<0.01; ***<0.001, ANOVA, Tukey-Kramer post hoc test. Source data are provided as a Source Data file.

animals did not adjust their strategy even based on missed food rewards. Across all animals, it appeared that there was a gradual transition in which reaches shifted towards location B on the timescale of days (Fig. 1g, i). We interpreted these results to mean that animals were initially in a rigid automatic state, and that a multiday process of reestablishing behavioral flexibility was required to consistently initiate trials to Location B; in turn, these data suggest that this switching process was not due to rapid within-session adaptation.

**Loss of consistent reach-locked M1 and DLS neural spiking during exploration.** What are the neural correlates of reestablishment of behavioral flexibility? We first focused on the task-

related firing of single neurons during the baseline periods and over the course of switching (Fig. 2a, b). We observed that for each animal there was usually one session in which the neural firing, when locked to reach onset, appeared to be quite variable from trial-to-trial. For example, when tracking M1 units over time, at sessions early to intermediate after the switch of the pellet to Location B, we saw a loss of consistent firing rate modulation aligned to reach onset (RO) across all trials in a session (Fig. 2a), despite rodents completing smooth, fast reaches (Fig. 1c, Supplementary Fig. 1f–j).

While M1 cortical dynamics in a reach-to-grasp task might become variable, it is still possible that striatal dynamics do not, especially given the similarity in reach "vigor" for reaches to A

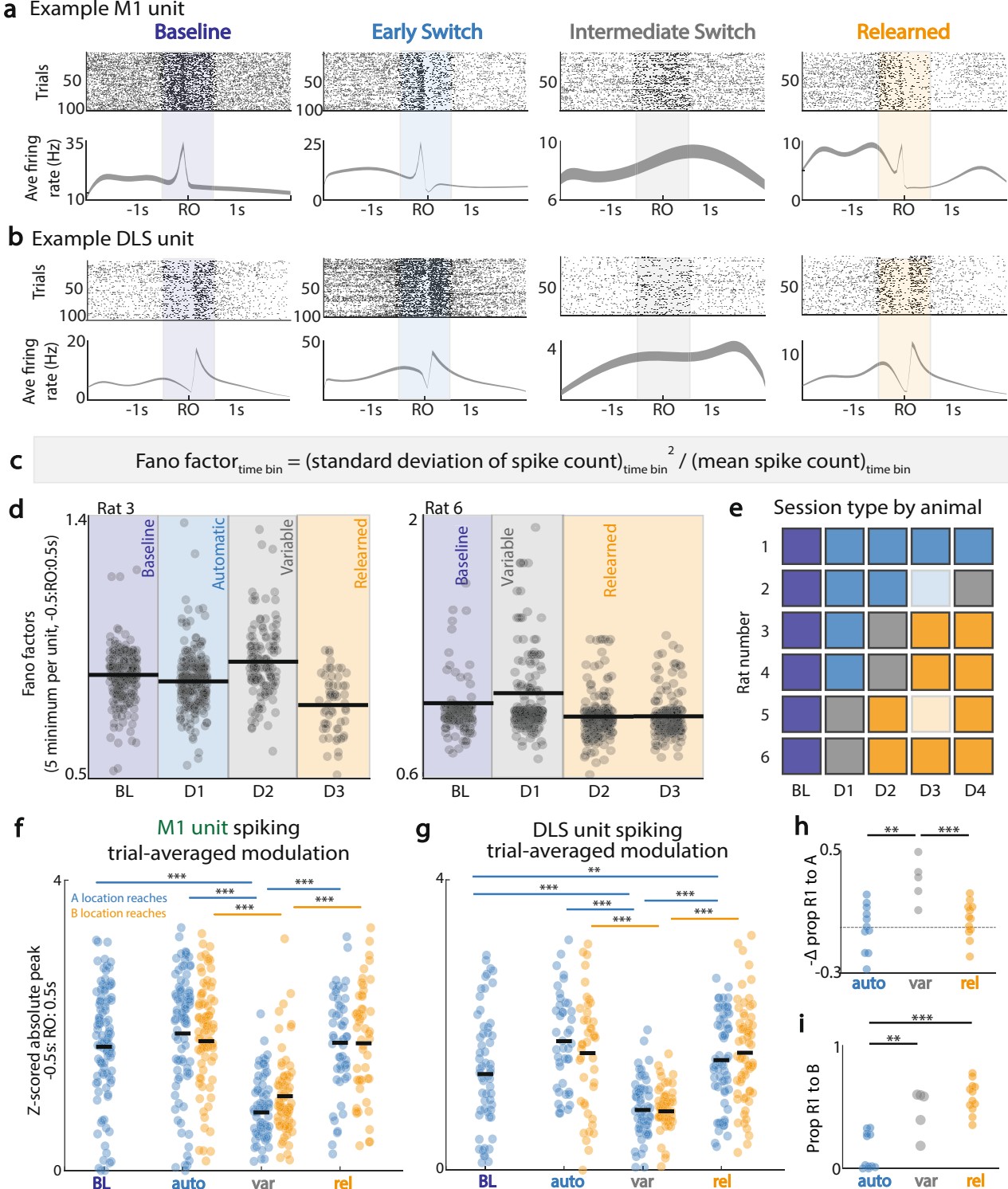

**Fig. 2 Breakdown of reach-locked M1 and DLS unit spiking during exploration. a** Top: Rasters for a single M1 unit† across sessions (same channel across days). Bottom: trial-averaged firing rate (mean ± SEM) for −2 s to 2 s around first reach onset. **b** Same as (**a**), for a DLS unit.† **c** Equation for determining Fano factor per time bin for each unit, per session. **d** Five minimum Fano factors per unit for −0.5 s to 0.5 s around first reach onset, across representative sessions, for two example animals. **e** Session type re-categorization based on Fano factors: indigo = baseline, blue = automatic (post-switch, no Fano factor change), gray = variable (maximum increase in Fano factor), orange = relearned. Light blue/light orange = two sessions not analyzed because of data corruption during recording. **f** M1 unit peak trial-averaged modulation around reach onset for trials with first reach to A (blue dots) and for trials with first reach to B (orange dots), across session types; peak modulation was calculated after z-scoring unit firing across the session. **g** Same as (**f**), for DLS units. **h** Change in proportion of first reach to A in a session, relative to proportion of first reach to A in previous session; grouped by session type across all animals, all sessions. **i** Proportion of first reach to B in a session, grouped by session type across all animals, all sessions. †Unit may not be the same across sessions. *<0.05, **<0.01, ***<0.001, linear mixed effects model, Bonferroni-corrected significance. Source data are provided as a Source Data file.

and B (Fig. 1c, Supplementary Fig. 1e–i). However, DLS units also exhibited a similar phenomenon as those in M1 (i.e., a drop in reliable modulation). Notably, such increases in variability were in tandem with M1, on the same sessions during which M1 spiking modulation decreased (Fig. 2a, b). Based on this observation, we quantified neural variability across both M1 and DLS using Fano factor of unit spiking activity at each time-bin throughout the trial (Fig. 2c). Fano factors, also known as indices of dispersion, are a measure of across-trial variability within a given time window, where high values indicate high trial-to-trial variability, and low values indicate low trial-to-trial variability[30,31]. We specifically measured Fano factors (i.e., variance/mean of unit firing at each time bin) for −0.5 s to 0.5 s around first reach onset and sub-selected the minimum 5 Fano factors per unit in a session, due to individual units having different "preferred" firing times relative to reach onset (Fig. 2d).

Using the Fano factor as a metric for normalized neural variance, we categorized behavioral sessions into four session types: (1) *Baseline* (BL), pellet at location A, reaching to A, low reach-related neural variance, as denoted by low neural variance and Fano factors; (2) *Automatic* (auto), pellet at Location B, reaching mostly to A, with low neural variance; (3) *Variable* (var), pellet at Location B, reaching to both A and B, and local maximum of Fano factors across units, with significant deviation from baseline minimum Fano factors; (4) *Relearned* (rel), pellet at Location B, reaching mostly to B, with low neural variance (see Methods, Spiking analysis, Determination of session type). Figure 2e illustrates how each of the daily sessions per animal were categorized based on these criteria. Importantly, subsequent analysis is based on this categorization of sessions, with one session of each type included per animal if available. Notably, Rat 1 did not demonstrate a *variable* or *relearned* session; this rat also had a flat decay curve in reaches to A, indicating strong perseverance to A. Rat 2 did not demonstrate a *relearned* session, although there were no sessions recorded after the *variable* session. Rats 5 and 6 did not demonstrate an *automatic* session and became *variable* early after the switch.

Interestingly, there was a significant drop in trial-averaged M1 and DLS unit modulation during *variable* sessions. Trial-averaged spiking modulation was calculated for each neuron in one session type per animal, by trial-averaging (50 ms bins) spiking activity aligned to reach onset and finding the trial average peak firing within a window of −0.5 to 0.5 s around reach onset. In M1, there was a significant drop in trial-averaged unit modulation on the *variable* session for trials with first reach to A (Fig. 2f, first reach to A BL: 1.71 ± 0.080 mean ± sem for here on; auto: 1.89 ± 0.093; var: 0.801 ± 0.057; rel: 1.76 ± 0.095; linear mixed effects model $p$ value, Bonferroni-corrected significance ($\alpha_{BC}$); lme model $p$ values, $\alpha_{BC} = 8.33e−3$, BL vs. var: 1.44e−14; auto vs. var: 2.66e−17; rel vs. var: 1.24e−15). Similarly, for trials with first reach to B, there was a drop in M1 reach-locked unit modulation on the *variable* session (Fig. 2f, first reach to B auto: 1.78 ± 0.088; var: 1.02 ± 0.064; rel: 1.75 ± 0.11; lme model $p$ values, $\alpha_{BC} = 0.0167$, auto vs. var: 4.48e−10; rel vs. var: 9.63e−10).

In DLS, the same pattern of the loss of trial-averaged reach-modulation on the *variable* session was observed for both trials with first reach to A and trials with first reach to B (Fig. 2g, first reach to A, BL: 1.32 ± 0.11; auto: 1.78 ± 0.099; var: 0.829 ± 0.057; rel: 1.52 ± 0.080; lme model $p$ values, $\alpha_{BC} = 8.33e−3$, BL vs. var: 3.46e−4; BL vs. rel: 3.60e−3; auto vs. var: 3.45e−12; rel vs. var: 3.19e−12; first reach to B, auto: 1.61 ± 0.12; var: 0.811 ± 0.046; rel: 1.62 ± 0.086; lme model $p$ values, $\alpha_{BC} = 0.0167$, auto vs. var: 2.29e−7; rel vs. var: 2.20e−14). Importantly, M1 and DLS average task firing rate does not change across session types (Supplementary Fig. 3a, b), and increased reach-related firing rate relative to non-reach periods per unit was preserved on the *variable* sessions

(Supplementary Fig. 3c; M1 units, paired $t$ test, $p = 1.17e−4$; DLS units, paired $t$ test, $p = 8.98e−6$).

We interpreted this to mean that during the identified *variable* session, there was increased variability in the neural spiking-behavior relationship; this was present in both M1 and DLS. Primarily for these sessions, there was a drop in the trial-averaged modulation; trial-averaged modulation was present for the other session types. Was there a link between when this variable neural activity occurred and the process of behavioral exploration? Interestingly, we found that the *variable* sessions were associated with a significantly larger drop in the proportion of first reach to Location A in a session as compared to the session preceding it (Fig. 2h, drop in proportion of first reach to A relative to preceding session (mean ± sem), auto: −0.0033 ± 0.046, var: 0.30 ± 0.064, rel: 0.047 ± 0.034; lme model $p$ values, $\alpha_{BC} = 0.0167$, auto vs. var: $p = 1.1e−3$; var vs. rel: $p = 9.4e−4$). Additionally, the *variable* session marked a timepoint when there was a transition to significantly more first reaches in a session to Location B (Fig. 2i, proportion of first reach to B (mean ± sem), auto: 0.14 ± 0.045, var: 0.47 ± 0.082, rel: 0.58 ± 0.037; lme model $p$ values, $\alpha_{BC} = 0.0167$, auto vs. var: 1.7e−3; auto vs. rel: 9.3e−8). Of note, while there was a gradual transition in reach endpoints to B during the variable session, the absolute spread of reach endpoints was not significantly different from baseline (Supplementary Fig. 2; average of reach endpoint x- and y-standard deviation (mean ± sem), baseline: 14.1 ± 2.58; auto: 14.5 ± 1.36; var: 11.4 ± 1.29; rel: 10.2 ± 0.692; lme model $p$ values: auto vs. BL: 0.68; var vs. BL: 0.068; rel vs. BL: 0.023). Thus, a loss of the previously stable neural spiking-behavior relationship across M1 and DLS and increased neural variability appears to coincide with both goal-directed behavioral exploration and the largest changes in behavior during the switch process.

Together, this analysis revealed that during behavioral exploration to Location B, there was an increase in trial-to-trial spiking variability in both M1 and DLS units. This was followed by a transition to a state with more consistent trial-to-trial activity in both regions. Interestingly, for each animal that demonstrated this *variable* state with a tandem loss of consistent across-trial M1-DLS firing, the number of sessions needed to reach this state appeared to be to be closely related to extent of training: rats with more training to Location A reached the *variable* state later after the pellet was switched to Location B (Fig. 2e, Supplementary Fig. 1c).

**Stability of 3-6 Hz M1-DLS coherence during variable state.** We also examined 3-6 Hz M1-DLS LFP coherence between session types; cross-area coherence in this band is known to evolve during initial skill learning[18]. We found, however, that there was no change in average trial coherence for 250 ms before and after reach onset between baseline A reaches and variable session B reaches. This stands in contrast with the increased coherence exhibited when first learning the reach-to-grasp skill[18] (Supplementary Fig. 4a, trial average 3–6 Hz M1-DLS coherence, −250 ms: reach onset: 250 ms, mean ± sem, BL = 0.492 ± 0.0017; var = 0.514 ± 0.0024). As our past work has linked the emergence of such coherence to faster and more consistent speed of movements[18], this finding of preserved coherence could reflect the additional observation that the speed of movements does not change in this paradigm during exploration and relearning.

**Loss of consistent grasp-locked M1 and DLS neural spiking during relearning.** Is it possible that single-unit modulation is more closely locked to grasping than reach onset? In other words, while there may be neural variability at reach onset in both M1 and DLS, does this subsequently normalize to consistent

patterning at first grasp (Supplementary Fig. 5a, b)? We found that there was a similarly significant increase in minimum Fano factors during the *variable* session in 4/5 animals. One animal did not have a significant increase; this is perhaps because of high baseline variability in firing at grasp onset. Using the same *variable* sessions as previously identified, we also calculated trial-averaged z-scored unit modulation in M1 and DLS during the period −0.5 s: first grasp: 0.5 s. In M1, there was a significant decrease in unit modulation from *baseline* and *automatic* to *variable* (Supplementary Fig. 5c, BL = 2.58 ± 5.9e−3; auto = 2.66 ± 9.3e−3; var = 2.3 ± 6.0e−3; *p* (BL-var) = 0.0012; *p* (auto-var) = 1.54e−5) and significant increase in unit modulation from *variable* to *relearned* (rel = 2.60 ± 9.8e−3; *p* (var-rel) = 1.12e−6). In DLS, there was a significant decrease in unit modulation from the *automatic* to *variable* sessions (Supplementary Fig. 5c, BL = 2.37 ± 0.12; auto = 2.76 ± 0.011; var = 2.10 ± 8.9e−3; *p* (BL-var) = 0.094; *p* (auto-var) = 2.30e−4), and significant increase from *variable* to *relearned* (rel = 2.70 ± 6.9e−3; *p* (var-rel) = 0.020). Thus, we interpret these findings to indicate a breakdown in both reach-related and grasp-related activity in M1 and DLS during behavioral exploration.

**Consistency of M1 and DLS temporal patterning**. A hallmark of consolidated skill control is consistency of M1 spatiotemporal activity patterns[6,9–11,14], which can be visualized by comparing a trial-averaged pattern to single-trial patterns (Fig. 3a, d). Previously we demonstrated a drop in trial-to-trial consistency of single-unit modulation relative to reach onset (Fig. 2). However, there could still be a weakly preserved spatiotemporal structure trial to trial. Consistency of single-trial pattern activation in a region can be assessed using a correlation between the single-trial spatiotemporal pattern and the trial-averaged spatiotemporal pattern ("template") for a session (Fig. 3a, b, d)[15]. Location A reach trials were compared to the template for all first reaches to A; the same process was completed for Location B trials in a session, as described in "Methods, Spiking analysis: Template matching".

Notably, there was a loss of reliable ensemble activation during the *variable* session (Fig. 3b). When comparing M1 spiking at first reach to A to the average neural template for such reaches, there was a drop in trial-template neural pattern consistency during the *variable* session as compared to the other sessions. Interestingly, there was a slight increase in neural pattern consistency from *baseline* to the *automatic* and *relearned* sessions (Fig. 3c, first reach to A (mean ± sem), BL: 0.267 ± 0.0096; auto: 0.342 ± 0.0085; var: −0.0480 ± 0.0094; rel: 0.356 ± 0.014; lme model *p* values, $\alpha_{BC}$ = 8.33e−3, BL vs. auto: 1.07e−4; BL vs. var: 4.34e−78; BL vs. rel: 9.32e−7; auto vs. var: 1.20e−100; rel vs. var: 8.42e−69). For first reaches to B, M1 spiking activity was similarly more temporally consistent for the *automatic* and *relearned* sessions as compared to the *variable* sessions (Fig. 3c, first reach to B (mean ± sem), auto: 0.280 ± 0.015; var: 0.0450 ± 0.013; rel: 0.0322 ± 0.0092; lme model *p* values, $\alpha_{BC}$ = 0.0167, auto vs. var: 3.34e−8; rel vs. var: 3.31e−55).

This same pattern of changes in trial-to-template correlation across session types was observed in DLS. Specifically, for first reaches to A and B, there was a decrease in spiking pattern consistency during the *variable* sessions, with a slight increase in pattern consistency for first reaches to A on the *automatic* and *relearned* sessions relative to *baseline* (Fig. 3e, first reach to A (mean ± sem), BL: 0.170 ± 0.012; auto: 0.184 ± 0.011; var: −0.0662 ± 0.011; rel: 0.240 ± 0.013; lme model *p* values, $\alpha_{BC}$ = 8.33e−3, BL vs. auto: 2.33e−3; BL vs. var: 3.23e−34; BL vs. rel: 9.54e−5; auto vs. var: 8.12e−20; rel vs. var: 4.04e−43; first reach to B (mean ± sem), auto: 0.193 ± 0.020; var: −0.0237 ± 0.0088; rel: 0.2208 ± 0.011; lme model *p* values, $\alpha_{BC}$ = 0.0167, auto vs. var:

2.05e−25; rel vs. var: 3.35e−55). This analysis emphasizes that not only does spiking return to reach-locked modulation (Fig. 2), but so does consistent spatiotemporal firing, indicating that reemergence of an optimal neural pattern follows the *variable* neural state in both M1 and DLS.

One question is whether units in either M1 or DLS are differentially tuned to the direction of reach. That is, are certain units differentially modulated at reach onset for reaches to Location A versus those to Location B? Surprisingly, when examining firing in both M1 and DLS units across sessions with consistent reaches to both A and B, only a small subset was differentially tuned in each region (Supplementary Fig. 6a, M1: 5–13%, DLS: 7–9%). This is consistent with recent work showing that condition-independent variance is significantly greater that condition-specific variance, such as the direction of reaches[32].

How does this variability in temporal patterning affect how M1 activity at time $t$ can predict subsequent M1 activity at time $t + 1$[9]? That is, do M1 dynamics on the *variable* day exhibit predictable linear dynamics[10]? To address this question, we fit an ordinary least squares linear dynamical system (LDS) model where M1 activity at the next timepoint is predicted by M1 activity at the previous timepoint. Using z-scored trial-averaged activity across session types, there was a significant drop in the variance explained ($R^2$) by the LDS model on the *variable* session relative to *baseline* and *automatic* sessions, indicating that on the *variable* session, there was little trial-aligned consistency in predictable M1 dynamics ($R^2$ across session types (mean ± sem: BL, 0.679 ± 0.055; auto, 0.753 ± 0.061; var, 0.287 ± 0.090; rel, 0.688 ± 0.046; lme model *p* values, $\alpha_{BC}$ = 8.33e−3, BL vs. var: 2.97e−3; auto vs. var: 4.62e−3). Thus, with increased variability of motor network patterns, there is a corresponding decrease in predictability of trial-aligned M1 dynamics.

**M1 and DLS single-trial population spiking activity loses temporal consistency with relearning**. Recent work has highlighted the notion that the total population level modulation[33] may be an important mode for M1 operation[34]. Specifically, reliably increased population sums—the sum of firing of an ensemble regardless of neuron identity—appears to be correlated with the reliable execution of reach-to-grasp movements[34]. In our case, while individual units may lose trial-averaged reach-specific modulation, it is possible that population sum firing in M1 and DLS is preserved, albeit with variable units participating across trials. This approach can help elucidate whether single-trial measures of firing that do not require trial averaging are still present. We examined M1 and DLS single-trial population spiking activity throughout the trial and across the different session types (Fig. 4a–c; Supplementary Fig. 3d).

On the *variable* session, there was indeed a drop in the peak of trial-averaged population activity, consistent with Fig. 2f, in a window of −0.5 s to 0.5 s around first reach onset, in M1 (first reach to A, BL: 2.12 ± 0.0029; auto: 2.33 ± 0.0026; var: 1.08 ± 0.0060; rel: 2.24 ± 0.0080; lme model *p* values, $\alpha_{BC}$ = 8.33e−3, BL vs. var: 6.87e−31; auto vs. var: 4.53e−33; rel vs. var: 4.72e−13; first reach to B, auto: 2.51 ± 0.0062; var: 1.59 ± 0.0045; rel: 1.98 ± 0.0038; lme model *p* values, $\alpha_{BC}$ = 0.0167, auto vs. var: 2.02e−5; rel vs. var: 4.27e−6). This was also the case for DLS (first reach to A, BL: 1.71 ± 0.0033; auto: 1.99 ± 0.0028; var: 1.28 ± 0.0053; rel: 1.94 ± 0.0078; lme model *p* values, $\alpha_{BC}$ = 8.33e−3, BL vs. auto: 1.97e−5; BL vs. var: 6.90e−3; auto vs. var: 3.14e−11; rel vs. var: 2.97e−9; first reach to B (mean ± sem), auto: 2.09 ± 0.0078; var: 1.34 ± 0.0041; rel: 2.05 ± 0.0034; lme model *p* values, $\alpha_{BC}$ = 0.0167, auto vs. var: 2.97e−11; rel vs. var: 7.05e−19).

However, when looking across the entire task period of −2s to 2 s around reach onset, there was little to no variation in session

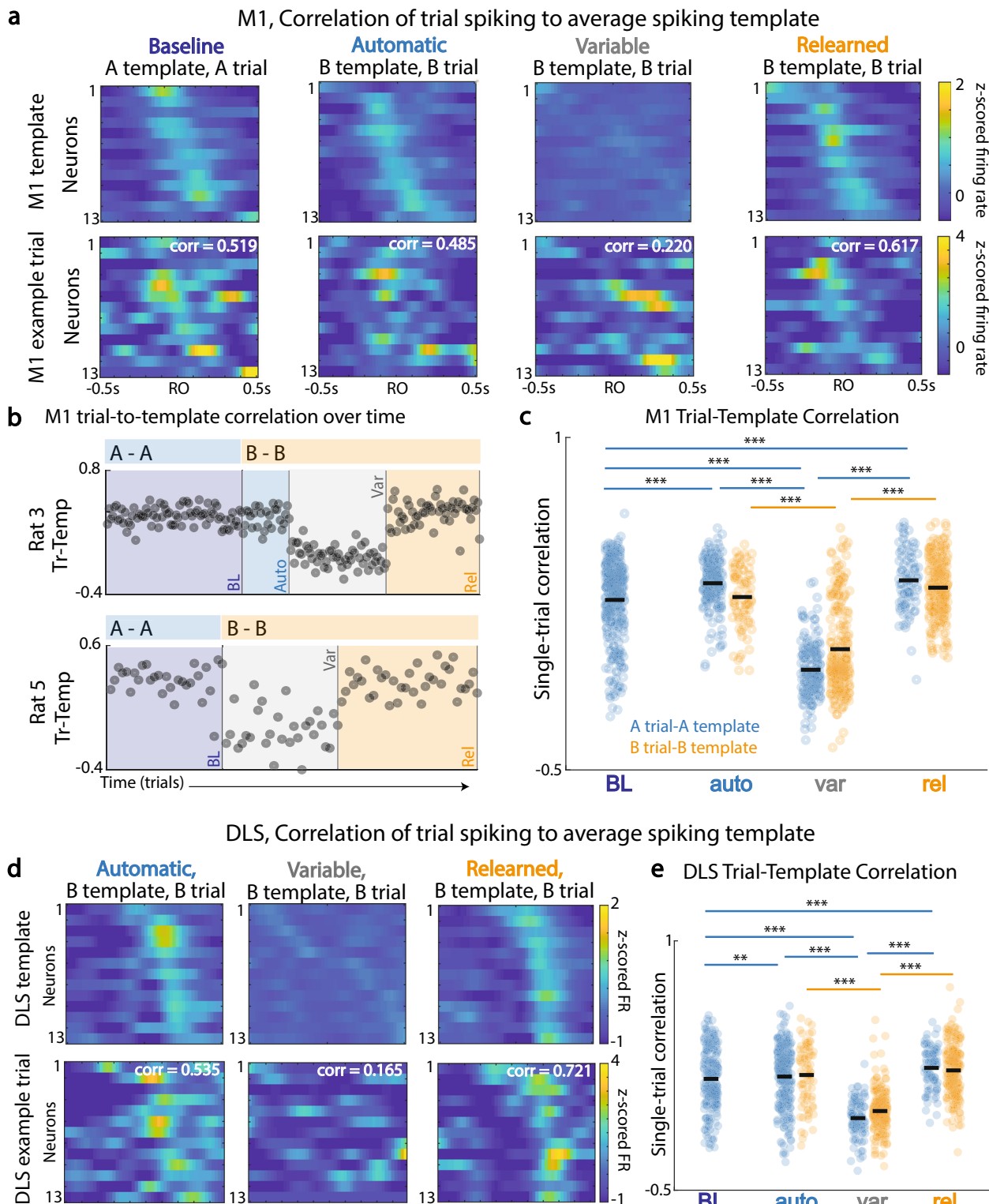

**Fig. 3 Changes in M1 and DLS single-trial ensemble reliability. a** Top: leave-one-out trial-averaged spiking pattern of M1 units for reach period, −0.5 s to 0.5 s around first reach onset, across session types. Bottom: left-out trial correlation with session template. **b** Trial-to-template correlation across trials, with A trial-to-template correlations shown for the baseline session, and B trial-to-template correlations shown for subsequent sessions. *Baseline* (BL), *automatic* (auto), *variable* (var), and *relearned* (rel). **c** Comparison of M1 trial-to-template correlations for all trials by session type. **d** Same as (**a**) for DLS units in example sessions. **e** DLS trial-to-template correlations for all trials by session type. *<0.05; **<0.01; ***<0.001, linear mixed effects model, Bonferroni-corrected significance. Source data are provided as a Source Data file.

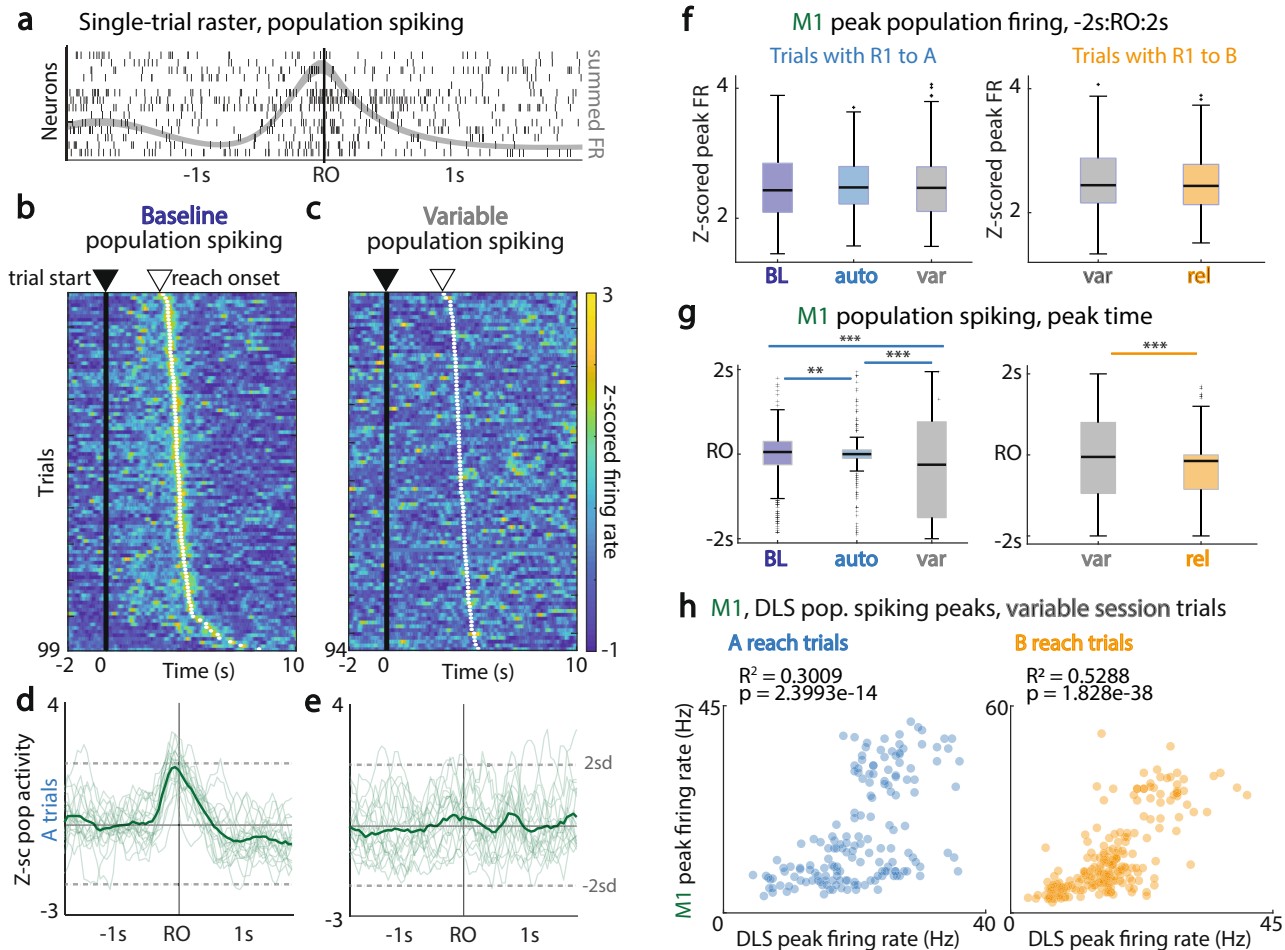

**Fig. 4 M1 single-trial population spiking activity loses temporal consistency with relearning. a** Example single trial raster plot with the population spiking average overlaid (summed spike counts, 50 ms time bins with spline fit). **b** M1 z-scored single-trial population spiking activity across a session; trials sorted by time from trial start to reach onset. **c** Same as (**a**), example animal *variable* session. **d** *Baseline* session z-scored population spiking in M1 around reach onset. **e** *Variable* session, example A reach trials. **f** Magnitude of trial peak population firing rate (z-scored sum of spiking over time) within −2 s to 2 s around first reach onset in M1 units for trials with first reach to A (left) and trials with first reach to B (right). R1 to A: BL, n = 291 trials; auto, n = 256 trials; var, n = 169 trials. R1 to B: var, n = 226 trials; rel, n = 214 trials. Data are presented as box plots with 25th, 50th, and 75th percentiles. **g** Time of population firing peak for the same trials, in same window as for (**f**). Data are presented as box plots with 25th, 50th, and 75th percentiles. **h** *Variable* session, covariation of M1-DLS firing across trials even with variable timing of peak. *<0.05; **<0.01; ***<0.001, Bartlett's test of homoscedasticity. Source data are provided as a  Source Data file.

single-trial peak population firing rate modulation in either M1 or DLS across session types. While Fig. 4d shows a consistent population sum peak that is locked to reach onset during a *baseline* session, the timing of the peaks is more variable during the *variable* session (Fig. 4e). The population sum peak across trials, across all animals, grouped by session type, is shown in Fig. 4f (M1, first reach to A, mean trial peak of M1 population firing ± sem, BL: 2.492 ± 0.0017; auto: 2.510 ± 0.0016; var: 2.528 ± 0.0033; rel: 2.508 ± 0.0048; first reach to B, auto: 2.575 ± 0.0055; var: 2.510 ± 0.0022; rel: 2.475 ± 0.0021; DLS: Supplementary Fig. 3f, first reach to A (mean trial peak of DLS population firing ± sem), BL: 2.527 ± 0.0017; auto: 2.433 ± 0.0018; var: 2.453 ± 0.0033; rel: 2.443 ± 0.0054; first reach to B, auto: 2.400 ± 0.0050; var: 2.509 ± 0.0023; rel: 2.406 ± 0.0022).

Is the drop in trial-averaged population spiking modulation then due to a shift in peak timing, as hinted by observation of trial firing (Fig. 4b–e)? While single-trial peak population firing rate throughout the task period remained relatively unchanged (Fig. 4f), for both A and B reaches, the time at which the peak of population firing occurred was significantly more spread out over the trial period

during the *variable* session, for both M1 and DLS. This was true in both M1 (Fig. 4g, first reach to A (mean time of peak trial M1 population firing in seconds relative to reach onset ± sem), BL: −0.0976 ± 0.0023; auto: −0.00960 ± 0.0022; var: −0.201 ± 0.0072; rel: −0.202 ± 0.0039; Bartlett's test *p* values, BL vs. auto: 4.50e−3; BL vs. var: 1.69e−19; auto vs. var: 1.05e−28; rel vs. var: 5.10e−11; first reach to B (mean ± sem), auto: −0.0250 ± 0.055; var: −0.0330 ± 0.049; rel: −0.328 ± 0.0035; Bartlett's test *p* values, auto vs. var: 6.50e−15; auto vs. rel: 2.73e−7; rel vs. var: 4.66e−6) and DLS (Supplementary Fig. 3g (top), first reach to A (mean ± sem), BL: 0.0684 ± 0.0032; auto: 0.0998 ± 0.0035; var: −0.199 ± 0.0070; rel: 0.0747 ± 0.0090; Bartlett's test *p* values, BL vs. var: 5.97e−4; auto vs. var: 2.26e−5; rel vs. var: 1.38e−4; Supplementary Fig. 3g (bottom), first reach to B (mean ± sem), auto: −0.0390 ± 0.0092; var: −0.0637 ± 0.0050; rel: −0.0671 ± 0.0038; Bartlett's test *p* values, auto vs. var: 4.67e−5; rel vs. var: 2.24e−6). Overall, this indicated that the timing consistency of M1 and DLS population spiking relative to behavior decreased during the *variable* session.

Especially given that M1 and DLS are monosynaptically connected structures, we next examined if, despite the loss of

consistent timing to the task, M1 and DLS population spiking modulation remained correlated relative to each other. Interestingly, despite the trial-to-trial inconsistency of the timing of peaks during the *variable* session, population spiking modulation was coordinated across structures. M1 and DLS exhibit correlated peak population firing rates (Fig. 4h; A reach trials, $R^2 = 0.301$, $p = 2.40e{-}14$; B reach trials, $R^2 = 0.529$, $p = 1.83e{-}38$), This indicates that during the *variable* session, while there was a loss of consistent trial-related activity in relation to reach onset, M1 and DLS had preserved tandem correlated population spiking. This might mean that M1-DLS are still coupled, albeit not in a precise task-related manner during the *variable* session.

**Changes in M1-DLS cross-area subspace modulation**. We next examined precisely how cross-area communication[35] between M1 and DLS changed during the *variable* sessions. We used canonical correlation analysis (CCA); CCA estimates linear combinations of neurons in M1 and DLS that represent maximally correlated activity across areas[26,36]. Through this method, axes of maximal correlation are identified for M1 and DLS (Fig. 5a, b), with subsequent projection of high-dimensional neural activity on these axes to examine shared signals in a lower dimensional space (Fig. 5c). This method also allows us to quantify cross-area coupling for baseline and task-related periods[26]. In particular, the models were fit on a broad task time period, $-2$s to $0.5$ s around reach onset, to additionally enable identification of when M1-DLS CCA subspace activity was most modulated.

To establish that CCA models of M1-DLS cross-area activity were behaviorally significant, we compared the $R^2$ of CCA models fit on actual data to the $R^2$ of CCA models fit on trial-shuffled data. Trial-shuffling as opposed to time-bin shuffling preserves variation due to trial structure, thus conservatively identifying M1-DLS correlations beyond what can be predicted from average trial activity. From this generated distribution of shuffled $R^2$ values, a CV from the true dataset was considered significant if its $R^2$ value exceeded the 95th percentile of the reference distribution (Fig. 5b). Most datasets had 1–3 significant CVs, demonstrating that CCA could identify shared low-dimensional activity across M1 and DLS; datasets that had no significant CVs were excluded from analyses.

To examine whether M1-DLS shared communication was task-relevant or reach-relevant, we compared M1 cross-area activity vs. DLS cross-area activity along the most significant CV in a session prior to reach onset and during reach for the different behavioral states (Fig. 5c, d). We then calculated the relative modulation index (RMI) of CCA activation per trial, to compare how reach-related cross-area activity was modulated over sessions (Fig. 5e). Of note, this comparison of modulation is predicated on whether the CCA model generalizability across days is comparable; that is, does CCA model $R^2$, a measure of the predictive power of the model, remain similar across days? We found that whether models were fit to trials with first reach to Location A or to Location B, $R^2$ values remained stable across session types (Supplementary Fig. 7a).

For trials with first reach to Location A, M1 cross-area activity was higher at the *automatic* state, as compared to the *baseline* and *variable* states (Fig. 5f, reach-modulation index, lme model $p$ values, $\alpha_{BC} = 0.0167$, BL vs. auto: 2.65e−12; auto vs. var: 5.19e−10). Cross-area activity in DLS exhibited a similar pattern, with reach-related cross-area activity higher at the *automatic* state, as compared to the *variable* state (Fig. 5f, reach-modulation index, lme model $p$ value, $\alpha_{BC} = 0.0167$, auto vs. var: 6.01e−4). Even for B reaches, this drop of reach-related cross-area activity on the *variable* sessions was pronounced (Fig. 5g), with moderate reemergence during relearning. M1 cross-area activity for trials

with first reach to Location B was higher at the *automatic* state as compared to the *variable* state, with increase in cross-area activity from the *variable* to *relearned* state (Fig. 5g, reach-modulation index, lme model $p$ values, $\alpha_{BC} = 0.0167$, auto vs. var: 1.34e−31; var vs. rel: 1.31e−9). Cross-area activity in DLS for reaches to B followed the same pattern, with decrease in activity from *automatic* to *variable* state and increase in activity from *variable* to *relearned* state (Fig. 5g, reach-modulation index, lme model $p$ values, $\alpha_{BC} = 0.0167$, auto vs. var: 7.75e−16; var vs. rel: 7.44e−8).

Could there be preserved reach-related CCA subspace modulation if the model is built on a reach-related epoch rather than a broader trial time window? We examined this question by building CCA models on a shorter time period, $-0.5$ s to $0.5$ s around reach onset, and comparing trial peak (maximum) subspace modulation at $-0.2$ s to $0.2$ s around reach onset across session types. Strikingly, across M1 and DLS we found lower peak trial subspace modulation for both A and B reaches during the *variable* session, as compared to the *automatic* state (Supplementary Fig. 7b, M1 peak trial subspace activation, lme model $p$ value, auto vs. var: 0.0015; DLS peak trial subspace activation, lme model $p$ value, auto vs. var: 5.9e−4) and *relearned* state, respectively (Supplementary Fig. 7c, M1 peak trial subspace activation, lme model $p$ value, var vs. rel: 0.014; DLS peak trial subspace activation, lme model $p$ value, var vs. rel: 0.0063).

Overall, this indicates that *variable* sessions are not only defined by increased task-related neural variability and a drop of reliable neural sequencing, but also a corresponding drop in task-related M1-DLS cross-area activity during reach (Fig. 5f, g). Notably, even while there was a significant M1-DLS subspace (i.e., correlated M1-DLS activity), there was less significant task-related modulation in the CCA space during the *variable* session in comparison to *automatic* or *relearned* sessions.

**Variable M1 cortical input drives variability of upper-limb movements**. Reliable patterning of spiking activity in M1 is widely believed to underlie the production of consistent skilled movements. We also found evidence for this in the *baseline, automatic* and *relearned* sessions. During the *variable* session, there was higher population firing than during non-reach periods, albeit with inconsistent patterning. Is it possible for there to be multiple firing patterns that could produce similar forelimb movements, and if so, does temporal variability enable constrained spatial variability? Intra-cortical microstimulation (ICMS) has long been a technique wherein one stimulation paradigm has been used to map movement from spatially varied stimulation sites across cortex. Few studies, however, have demonstrated how timing of impulse delivery may change output movement, with none controlling for amount of charge over time. Further, it is entirely possible that downstream structures—red nucleus, spinal cord—can filter out the variability in M1 firing patterns and enable more consistent movements.

To directly test the hypothesis of whether trial-to-trial variability in M1 neural activity could drive downstream structures to produce kinematic variability, we applied ICMS in M1 under ketamine anesthesia (Fig. 6a, also see Methods). ICMS is known to activate descending fibers from M1 to the motor periphery[37,38]. Thus, ICMS allows us to directly test how movement-related structures downstream from M1 might respond to variability in M1 patterning. We specifically compared two stimulation patterns: (1) patterned burst stimulation (PB), with consistently timed 333 Hz triplet pulses, delivered at 10 Hz[26,39] and (2) random single pulse stimulation (RP), with randomly delivered single pulses at 30 Hz (Fig. 6b, c). Each "session" was comprised of 2 s of a given stimulation paradigm that was delivered multiple times per animal with an intertrial

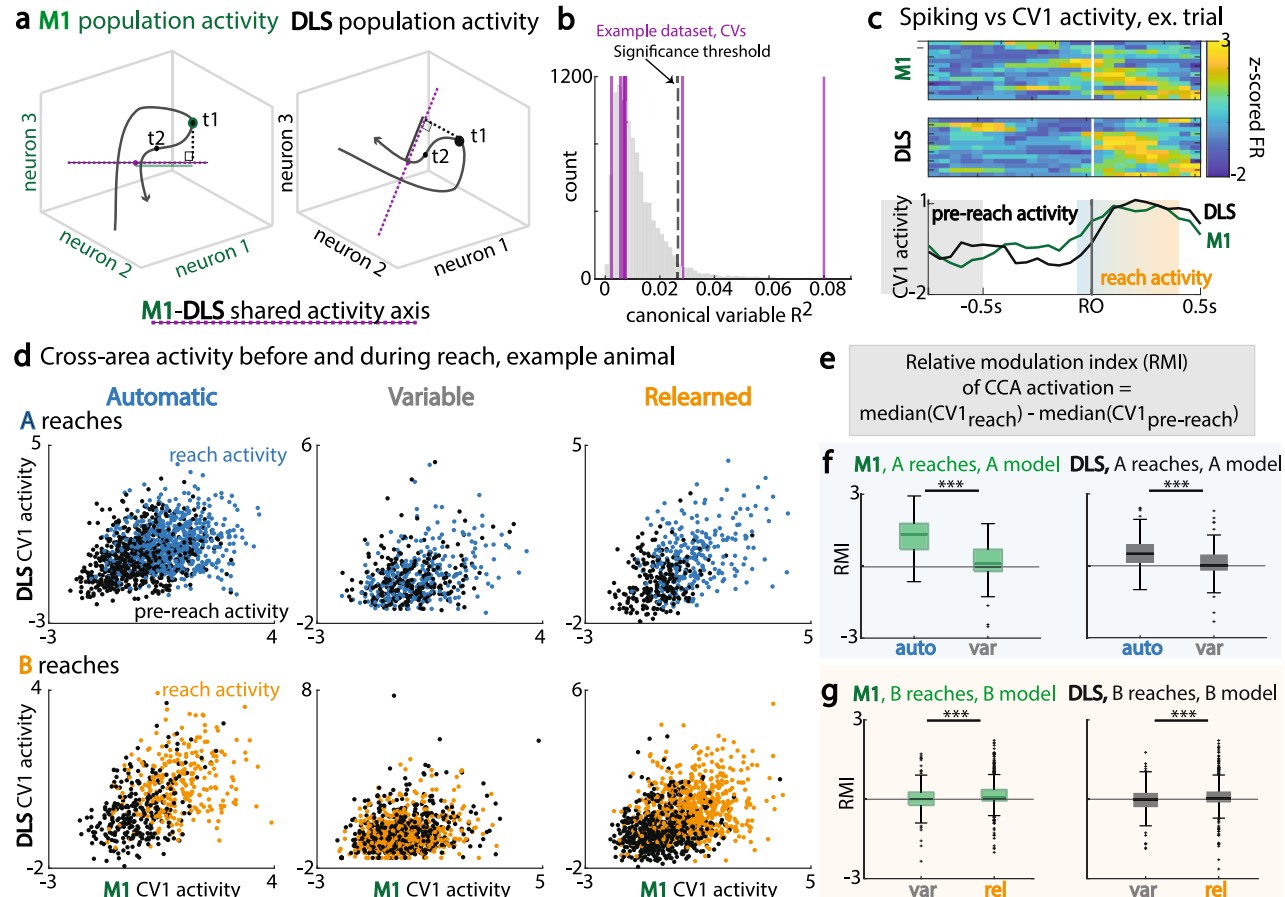

**Fig. 5 Changes in M1-DLS cross-area subspace modulation. a** Canonical correlation analysis (CCA): identification of axis with maximal correlation between M1 and DLS activity (purple dotted line), with example value of projected population activity at time t1 (solid line on shared axis) for M1 population activity (left, green) and DLS population activity (right, black). **b** Example bootstrap shuffled distribution of $R^2$ values for identification of significant canonical variable (CV) axes. **c** Top: single-trial M1 spiking activity. Middle: single-trial DLS spiking activity. Bottom: corresponding single-trial M1 (green) and DLS (black) activations along CV1. Pre-reach period (gray) is −1 to −0.5 s before first reach onset. Reach period (blue-orange) is −0.1 s to 0.4 s around first reach onset. **d** M1 and DLS cross-area activity before (black) and during (color) reach, for A trials (top row) and B trials (bottom row) across session types for example animal. **e** Equation for relative modulation index (RMI) of CCA activation, comparing reach and pre-reach periods. **f** RMI for trials with first reach to A, for M1 activations (left, green) and DLS activations (right, green), across automatic and variable sessions. R1 to A, M1 and DLS RMI: auto, n = 220 trials; var, n = 95 trials. Data are presented as box plots with 25th, 50th, and 75th percentiles. **g** Same as (**f**), for trials with first reach to B, across *variable* and *relearned* sessions. R1 to B, M1 and DLS RMI: var, n = 150 trials; rel, n = 464 trials. Data are presented as box plots with 25th, 50th, and 75th percentiles. *<0.05; **<0.01; ***<0.001, linear mixed effects model, Bonferroni-corrected significance. Source data are provided as a Source Data file.

period. Thus, while the total number of pulses were constant (60 pulses/session), the main difference was temporal consistency of the pulse pattern. Channels that elicited forelimb movement with ICMS were identified and used for subsequent sessions. Within each animal, multiple sessions of either patterned burst stimulation or random pulse stimulation were delivered. We recorded video of each stimulation session, marking digits 3 (D3), digit 4 (D4), and the center of the paw (CP) using DeepLabCut[40] for trajectory analysis (Fig. 6d).

Given that reaching movements last a couple of hundred milliseconds during awake behavior, we used a sliding 300 ms window across each session to identify "trials" (there were multiple movement trials evoked per session). More specifically, if a given 300 ms window in either stimulation conditions had a comparable number of pulses and was linked to a movement, this was used as a "trial" for each condition. Trial identification was limited to the first 1 s within a session, due to limited forelimb movement in the second half of the session, possibly due to fatigue. Subsequently, endpoint locations for each trial were

defined as the x- and y-coordinates at the end of the 300 ms window relative to spatial location at the start of the trial window.

Between the two stimulation types, we observed within-animal differences in endpoint location across three animals (Fig. 6e). For each marked point across animals, there was a significant difference in distribution of endpoints between patterned burst and random pulse stimulation, with patterned burst stimulation resulting in more consistent endpoint locations as seen by the skewed versus more uniform distributions, respectively (Fig. 6e, Kolmogorov-Smirnov test for two samples: Rat 1: $D3_x$, p value = 6.6e−5; $D4_x$, p value = 1.5e−3; $CP_x$, p value = 2.0e−13; $D3_y$, p value = 5.0e−4; $D4_y$, p value = 0.016; $CP_y$, p value = 1.1e−11; Rat 2: $D3_x$, p value = 0.010; $D4_x$, p value = 1.6e−3; $CP_x$, p value = 1.8e−11; $D3_y$, p value = 5.0e−3; $D4_y$, p value = 2.1e−3; $CP_y$, p value = 1.8e−11; Rat 3: $D3_x$, p value = 6.6e−3; $D4_x$, p value = 6.1e−4; $CP_x$, p value = 0.0112; $D3_y$, p value = 9.6e−7; $D4_y$, p value = 7.8e−6; $CP_y$, p value = 0.022). We interpret this finding to indicate that consistent recruitment of downstream regions by single-site stimulation in

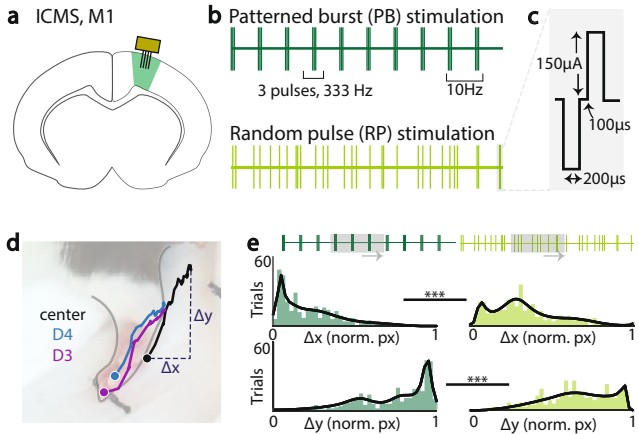

**Fig. 6 The role stimulation variability in driving variability of upper-limb movements. a** Location of stimulation, via microwire array to M1 for identification of regions eliciting forelimb movement. **b** Patterns of stimulation delivery, with balance of total charge delivered. Patterned burst stimulation (top, PB) consisted of 333 Hz triplets, at 10 Hz triplet frequency. Random pulse stimulation (bottom, RP) consisted of single pulses, randomly timed at 30 Hz. Stimulation was delivered for 1 s per session, for a total of 30 pulses per session. **c** Stimulation pulse parameters, with 200 µA biphasic pulse, 200 µs per phase with 100 µs inter-phase interval. **d** Lateral view of paw: limb markers labeled for supervised kinematic tracking: tip of digit 4 (D4, blue), tip of digit 3 (D3, purple), and center of paw (center, black). Endpoint location of each trial ($\Delta x$, $\Delta y$) was calculated relative to start location in trial window and normalized by animal across stimulation conditions. **e** Example animal histograms of normalized $\Delta x$ (top) and $\Delta y$ (bottom) for center of paw across patterned burst stimulation trials (left) and random pulse stimulation trials (right). Gray boxes show an example 300 ms window with identical number of pulses. Kolmogorov-Smirnov test for two samples in this example animal, center of paw, $p = 1.8e{-}11$; *<0.05; **<0.01; ***<0.001, Kolmogorov-Smirnov test for two samples. Source data are provided as a Source Data file.

M1 with a patterned burst stimulation produces consistent forelimb behavior. In contrast, variable stimulation of down-stream networks produced constrained—still producing forward forelimb movement—yet variable endpoint control.

## Discussion

The main goal of our study was to understand how behavioral flexibility is established in the context of automatic performance of a complex motor skill; a closely related goal was to understand how M1 in tandem with DLS can account for flexible transitions between behaviorally rigid and exploratory states. Automaticity is a general term encompassing movement that is performed with little cognitive effort and resilience to intruding actions[21,22,41]. A growing body of literature has equated recruitment of the striatum as a neurophysiological correlate of automaticity[17,22,42]. Thus, as expected, we found that extensive practice on the reach to grasp task resulted in stable performance and reliable ensemble patterns in both M1 and DLS. As outlined in the introduction, it remained unclear how M1 and DLS might change in response to an alteration of the pellet location and large consistent behavioral errors. We found that in our *baseline* state of stable performance and with consistent recruitment of M1 and DLS, animals were in an automatic and rigid state, as evidenced by subsequent absence of within-session adaptation. Establishing behavioral flexibility of reach and grasp to the new location was observed to be a multi-day process. Notably, there was a transition to a *variable* neural state, both at the level of single neurons and ensemble patterns, that closely coincided with behavioral exploration; this was then

followed by successful convergence on a stable neural pattern with relearning. We also observed a significant drop in CCA subspace modulation between M1-DLS during this exploration, followed by significant restoration with relearning.

Our observation of tandem variability in both M1 and DLS also raised the important question of how M1 can still drive movements with such variability of neural patterns. This is particularly important because of the growing body of literature that almost exclusively views M1 as a robust and reliable pattern generator[9]. Thus, we tested whether the specific patterning in M1 could drive either precise or variable movements. In contrast to reliable ICMS patterned bursts, pulses delivered randomly resulted in greater endpoint variability, comprising two differing movement output patterns with overall conserved electrical input. Together, this suggests that the state of M1-DLS task-related coupling, either strongly coordinated in a shared subspace or weakly coordinated, may reflect M1 control modes: either a reliable pattern generator for automatic and precise movements or a variable controller for behavioral exploration.

**Relation of a motor skill to habits**. There have been recent debates about the differences between a motor skill and a habit[43,44]; automaticity, or the unconscious performance of an action, could be applied to both[2,17,43]. Perhaps one important difference is the focus on specific qualities of the motor action being performed for a skill, versus a focus on lack of sensitivity to devaluation for a habit. Classically, establishment of habit has been promoted via random-interval delivery of reward, in contrast with random-ratio delivery of reward for goal-directed movements[45,46]. Moreover, specific perturbations are performed to assess the characteristics of devaluation[21]. The focus of our study, however, was solely on the characteristics of the action being performed and not the process of devaluation, per se. For example, because of our focus on the action itself, we could track changes in reach endpoint differences during the process of relearning. Moreover, while there are likely to be similarities in the involvement of the striatum in both skill acquisition and habit formation, an important difference may be in how M1 activity changes for the particular action (i.e., reach-to-grasp versus lever pressing[40] versus T-maze navigation[47]). In contrast to more complex skilled actions, habits are often assessed using simpler motor actions which may have lesser contribution of M1 activity to movement output. For example, it is less clear how important M1 is for highly consolidated lever pressing, an action that is commonly used to study habits[48]. Thus, our goal was to focus on detailed assessments of a complex M1 and striatum dependent reach-to-grasp action[18,27,49], characterize automatic versus exploratory modes of control, and detail how M1-DLS represented these two control modes.

How is habit established and broken for simple motor actions? This has been most rigorously studied using the T-maze task. With learning, DLS neurons preferentially entrain to ~5 Hz oscillations and demonstrate robust task-specific activity[47]. With habit formation, there is a shift in ventromedial striatum spiking from 70–90 Hz (high gamma) bursts to 15–28 Hz (beta) bursts, with local and widespread spike-field synchrony in the high gamma and beta bursts, respectively[50]. In DLS, habit formation results in decreased firing in non-rewarded trials, and loss of error-related signaling. Intriguingly, with reward devaluation and subsequent reward return, there was a corresponding absence of post-goal activity followed by return to early learning firing patterns, lending evidence towards striatal plasticity and return to a partially variable neural state[51]. Thus, from the perspective of DLS activity, there are clear similarities with our results. However, because of our focus on the detailed kinematics of the action itself

and the role of changes in M1 modes (neither of which is known for the studies above), we can provide insight into how M1 is able to drive automatic versus exploratory behavior for a complex motor skill.

**How can M1 neural variability aid behavioral exploration?** Our results indicate that M1 can flexibly transition between predictable and variable modes. Our reach-to-grasp behavior requires coordinated control of both a gross reaching and fine grasping movement[18,25,28]. There are multiple descending pathways from M1 that might be able to support such an action. For example, it is well known that the red nucleus, at least in rodents, is important for grasping actions[52]. Other brainstem nuclei can also support reaching movements[53]. Consistent with a large body of literature, our results suggest that M1 input might recruit these parallel streams to drive reach and grasping actions. Our results add insight that the variability of such actions may be the result of active regulation of variability in M1 patterns. For example, we found that variability of M1 ICMS pulses resulted in greater endpoint variability. More broadly, our results suggest that M1 patterning can be either reliable and consistent or variable; moreover, M1 variability appears to be actively accompanied by variability in striatal activity. It is likely that downstream regions can 'decode' such differences in M1 variability while still enabling coordinated movements. Importantly, each particular movement during the *variable* session was performed in a fast and smooth manner, with M1-DLS variability only apparent across this subset of all trials during the relearning paradigm. This further highlights the importance of alternate regions downstream of M1, including the spinal cord[54], in generating coordinated movements. Without such supporting structures, loss of the spiking-behavior relationship and ensemble dynamics in M1 would likely not result in coordinated muscle activations and coordinated reach to grasp actions.

There are at least two parallel projection systems from M1, the intratelencephalic (IT) type neuron pathway, which projects to cortical and basal ganglia regions and the pyramidal tract (PT) type neuron pathway, which projects down to thalamus and brainstem with some collaterals to striatum[55,56]. Thus, one hypothesis that could reconcile how we observe fast movements during behavioral exploration on the *variable* session is that the IT and PT pathways independently regulate activity patterns. For example, in an automatic state, both IT and PT pathways could be following predictable dynamics enabling consistent execution of the consolidated motor skill. However, in the behaviorally exploratory state, IT pathway firing in M1 and DLS may be disrupted to enable manifold expansion and drive behavioral exploration, whereas PT pathway activity in M1 and downstream brainstem nuclei remains relatively preserved to support the execution of a fast reach-to-grasp action, in addition to constraining behavioral exploration introduced by the variable IT patterning. With further expansion of cell-type specific electrophysiology techniques, future studies could detail the individual contributions of the IT and PT pathways to establishment of behavioral flexibility and generalizability in a previously narrowly learned skill.

**How might motor noise contribute to modifying a skill in response to errors?** In our data we observed that some animals learned the second location more quickly than others. One possibility for these individual differences could be explained by whether the second pellet location was within the "motor noise" of the first pellet location that was originally trained. Particularly in songbirds, it has been demonstrated that even if a skill is "crystallized", adaptation within motor noise can occur rapidly

and is supported by the analogous basal ganglia structures[57,58]. This phenomenon has also been demonstrated behaviorally in humans, wherein increased baseline motor noise predicts subsequent rate of adaptation in both reward-based and error-based learning tasks[59]. This might suggest that even for "automatic states", as defined by evidence for strong task engagement of the DLS and increased coupling between M1 and DLS, there may be differences in the extent of motor noise. As suggested by our finding of a correlation between rate of learning the new location and the number of trials over task training (Supplementary Fig. 1c), this might be related to the extent of training and consolidation. Interestingly, our past study has found a role for sleep in driving increases in the direct coupling between M1 and DLS[24]; thus, it is possible that an interplay between the task exposure and long-term consolidation processes could regulate the extent of flexibility and motor noise in an automatic state.

**What mechanisms might underlie changes in M1-DLS variability?** What might be the behavioral advantage of establishing a variable state in M1 and DLS? One can begin with addressing the corollary, of why a refined and perhaps rigid state is produced in the first place. With establishment of optimal behavioral patterns for consistent reward, neural ensembles are similarly refined and strengthened via dopamine-dependent signaling at corticostriatal synapses[17,24,60]. However, when these behavioral and neural patterns are insufficient for reward, new ensembles must be generated for exploration. Thus, this "relearning" process of switching from previously rewarded ensemble activity to variable firing patterns likely involves a global network state shift, within both cognitive and motor circuits, towards generation of newer, more optimal ensemble activity for reward. Indeed, recent work in mice has demonstrated that inter-trial signaling in M1 is necessary for updating a reach-to-grasp behavior in response to motor error caused by change in pellet location[61]. Unanswered in our study is whether the newly developed ensemble activity is a fully novel pattern, or rather a minor adaptation of the previously learned patterning, which could be answered with continuous tracking of identical neurons. Additionally, future research with multi-area recording across associative and sensorimotor networks simultaneously, with online inactivation of each region independently, will likely elucidate more broadly how this switch from a rigid to flexible behavioral state occurs.

This study finds that a significant shared CCA subspace between M1 and DLS can be identified across a relearning paradigm; however, it is unknown whether M1 neural variability is solely driving DLS neural variability during the behaviorally exploratory state. While M1 and DLS are monosynaptically coupled, DLS also receives inputs from multiple other cortical and subcortical areas. Thus, a third structure, such as the secondary motor cortex[26] or prefrontal cortex[62], both of which have projections to M1 and DLS, may be coordinating this simultaneous establishment of neural variability. Simultaneous recording in these four regions during early motor skill learning, automatic performance, and introduction of a second motor skill, would more definitively address directionality of M1-DLS coordinated activity during this task. It is also worth examining the specific contributions of M1 and DLS to movement control. Lesions of M1 after a well-learned gross motor skill, without a fine motor prehension component, has been demonstrated to have no effect on action execution[48], hinting that downstream structures such as DLS may be driving gross forelimb movement, such as reaching, rather than M1. However, our past work has indicated that adding a prehension component to the skill makes it continue to require M1[18]. Moreover, we also saw that DLS task-related activity was lost during the *variable* session. This further

suggests that our observed DLS activity was not solely driving the reaching action, especially as we observed that the animals made directed and fast reaching actions, albeit without the grasping component. Notably, we also found that, even for the *variable* sessions, there was a significant correlation between M1 and DLS peak firing—the timing of such peaks was not reliably task locked (Fig. 4). Based on the recent finding that single-trial neural responses may be dominated by unstructured non-task movements[63], it is possible that such movements underlie the correlation between M1 and DLS during *variable* sessions. Alternatively, the correlated activity may be 'internal' to the nervous system, i.e., not movement-related.

**Which brain regions beyond M1 and DLS could support movement generation?** Primary motor cortex has been recently described as a reliable pattern generator for producing consistent patterns to drive downstream structures[10,13,15]. While neural activity could obey predictable dynamics for execution of well-learned movements, it is apparent from neural variability in early motor learning[6] and now relearning that other nodes in the larger brain network can support movement execution outside of the well-learned skill state. What are specific regions other than M1 and DLS directly implicated in movement pattern generation, from cortical regions through peripheral neurons controlling muscle activation? The prefrontal cortex (PFC) along with dorsomedial striatum (DMS) have been heavily implicated in associative learning across species[62,64]. Thus, with reestablishment of behavioral flexibility, there could be a shift in pattern generation from M1-DLS to PFC-DMS for modification of directionality in reaching. Due to recurrent networks involving each of these node pairs, a parallel hypothesis is that specific patterns seen in a well-learned M1-DLS state are a byproduct of strong recurrent network activation. In turn, with exploratory behavior, there could be less consistent neural patterning in PFC-DMS as well, but increased firing rates are sufficient to drive downstream motor patterns with constrained variability, as detailed earlier by less variant motor nodes such as brainstem nuclei and the spinal cord.

**Summary and proposed model.** We propose the following model for neural control of flexible movement generation, especially to place our findings in a larger context. First, with learning of a motor skill, co-activation of associative and sensorimotor loops establishes a refined and consolidated neural patterning, or manifold. However, if the skill is too narrow, as in our case, a broadening of the learned skill appears to occur via substantial behavioral exploration followed by convergence to a new strategy; this is contrasted from rapid behavioral adaptation[3,23]. The tandem increase in M1-DLS variability may support behavioral exploration to support greater generalization of the skill. It is possible that the intrinsic motor manifold may need to be altered to support these new movements. How might this relate to studies of larger repertoires of skilled movements? It is quite possible that if a skill is initially learned in a more general manner (e.g., reaching to multiple locations), the emergent neural manifold may be able to support a more generalizable skill[3]. In that case, we might predict more rapid behavioral adaptation[7]. It is possible that this is evident for studies in non-human primates and humans performing center-out reaching and reach-to-grasp actions[9,32,65,66]. However, it also remains unclear how to precisely relate rodent based findings on automaticity and habits to humans[44]. Interestingly, even after relearning, we noticed that animals continued to occasionally reach to the old location. This may be analogous to the notion of "slips of action" seen in humans, where the wrong action is selected when switching between tasks[44].

## Methods

**Animal care.** All procedures were in accordance with protocols approved by the Institutional Animal Care and Use Committee at the San Francisco Veterans Affairs Medical Center. Adult male Long-Evans rats between 3 and 6 months old ($n = 9$, 300–500 g; Charles River Laboratories) were housed in a controlled temperature room with a 12 h light/12 h dark cycle. All experiments were conducted during the light cycle.

**Surgery.** All surgeries were performed using sterile surgical technique under 2% isoflurane (5% at induction). Prior to implantation, forelimb preference was identified for contralateral electrode implantation (See "Methods, behavioral training"). Surgery involved exposure and cleaning of the skull, preparation of the skull surface (using cyanoacrylate) and implantation of skull screws on the perimeter for headstage stability. Reference screws were implanted posterior to lambda, contralateral to neural recordings. Ground screws were implanted posterior to lambda, ipsilateral to neural recordings. Craniotomy and durectomy were performed, followed by implantation of neural probes. Neural probes (32- or 64-channel 33um polyimide-coated tungsten microwire electrode arrays (Tucker-Davis Technologies)) were implanted in the forelimb area of M1 (centered at 3.5 mm lateral and 0.5 mm anterior to bregma; layer V at a depth of 1.5 mm) and DLS (centered at 4 mm lateral and 0.5 mm anterior to bregma; at a depth of 4.5 mm). The final location of the electrodes was confirmed by electrolytic lesion. Post-operative recovery regimen included the administration of 0.02 mg/kg buprenorphine for 2 days, and 0.2 mg/kg meloxicam, 0.5 mg/kg dexamethasone and 15 mg/kg trimethoprim-sulfadiazine for 5 days. All animals were allowed to recover for 1 week prior to further behavioral training.

**Histology.** Rats were anesthetized with isoflurane and transcardially perfused with 0.9% sodium chloride, followed by 4% paraformaldehyde. The harvested brains were then postfixed in 4% paraformaldehyde for 24 h and immersed in 20% sucrose for 2 days prior to drop freezing. Coronal cryostat sections (40 μm thickness) were mounted with permount solution (Fisher Scientific) on super-frosted coated slides (Fisher Scientific). Microscope images of the whole section were taken by a Zeiss microscope.

**Behavioral training.** Rats naive to motor training were first assessed for forelimb preference: ~10–20 pellets were placed in front of the animal, with preference defined by the limb which reached to the pellets the most. 4 out of 6 animals then underwent surgery for electrode implantation followed by a recovery period. Rats were then trained within an automated behavior box to perform dexterous reach-to-grasp movements[29] to a single location A for at least 1000 trials. 2 out of 6 animals were trained similarly, with electrode implantation 1 week after training the reach-to-grasp motor skill to A. Following surgical recovery, these 2 animals were retrained to baseline level of reach to A. Overall, this initial behavioral training required minimal user intervention, as the automated reach-box was controlled by custom MATLAB R2018B (MathWorks) scripts and an Arduino microcontroller.

Different rats had different "preferred" reaching locations with regard to cross-directionality and amplitude (i.e., a right paw-reaching rat could reach either straight ahead or across to the left without encountering the wall, with sufficient distance away from the wall such that reaching was encouraged over licking); this first location was defined as 'A'. 'B' was a location that was one pellet arm width, 1 cm, over from reach location A, either towards the center or laterally, with the same distance away from the center of the slit in the wall as A was (i.e., different angle from midline, same reach amplitude).

Preceding each trial, a pellet was dispensed on the end of an arm with pellet holder groove and moved to the pre-programmed location. Each trial was then cued by a tone, followed by opening of the door, allowing access to the pellet. Animals needed to then reach through a slit, grasp, and retrieve the pellet within 15 s. An IR sensor centered over the pellet was used to detect when there was no longer a pellet in the groove, indicating the trial was over; the door then closed. Each animal was trained to plateau performance of reaching to A (~100–150 trials per day, range = 9–16 days), prior to pellet location being switched to a second location, B, as described in the previous paragraph. Animals were then trained on the second pellet location B for 4–6 days (~100 trials per day). Of note, rodent detection of pellet location has been demonstrated to be most likely via olfaction, as opposed to visual discrimination[67].

**Behavioral analysis.** For 2/6 animals, behavior was video recorded using a side-view camera. For the remaining 4/6 animals, behavior was recorded using both top-down and side-view cameras. Three types of cameras were used: Microsoft LifeCams, which captured videos at 30 Hz; Basler cameras, which captured videos at 75 Hz; and Point Gray/FLIR cameras, which captured videos at 75–100 Hz. Reach trajectories were captured from video using DeepLabCut[40] v2.2 to track the center of the rat's paw. Reach trajectories consisted of paw trajectory from each reach onset to subsequent grasping motion that occurred beyond the slit. Reach videos and trajectories were viewed and scored to obtain trial success, reach type (low amplitude, endpoint at old location A, endpoint at new location B), and timepoints for reach onset, pellet touch, grasp onset, and retract onset. For

consistent comparisons across sessions, the first 100 trials in each session were analyzed. Reach onset (RO) was defined as initiation of forward displacement of the paw after the paw has completely rotated from flexion to extension.

To characterize motor performance, we quantified reaction time, reach duration, pellet retrieval success for each trial, and location of reach endpoints both within and across trials. Reaction time was defined as the time taken from when the door opened for the start of the trial to when the rat began to reach, combining both attentional and cue-related motion behaviors (Supplementary Fig. 1g). Reach duration was defined as the time from first reach onset to first grasp onset (Supplementary Fig. 1h). Percent reach success was defined as the percent of trials on which a pellet was successfully retrieved out of total trials with a full amplitude reach within a session. Low-amplitude reaches were those in which the center of the paw reached past the slit but digits did not reach the vertical plane where the pellet was located. Reaches to the new location B were defined as those where at least half of the paw covered the pellet on grasp. All other full amplitude reaches were classified as reaches to old location A.

## Electrophysiology data collection

We recorded extracellular neural activity, including units and LFP, using an RZ2 system (Tucker-Davis Technologies). Spike data was sampled at 24,414 Hz; LFP data was sampled at either 1017 Hz ($n = 2$) or 24,414 Hz ($n = 4$). Snippets of data that crossed a high signal-to-noise threshold (at least 4 standard deviations away from the mean) were deemed spiking events; time stamps and peak-aligned waveforms were stored for any event that crossed the threshold. Spike sorting was then performed using Offline Sorter v.4.3.0 (Plexon) with a k-means-based clustering method followed by manual inspection. Spikes were sorted separately for each session. We accepted units based on waveform shape, clear cluster boundering in principal component space, and 99.0% of detected events with an ISI > 2 ms. Clusters interpreted to be single units or multi-units were kept for analysis; those determined to be noise were discarded.

## Neural data analysis

Analyses were conducted using a combination of custom-written scripts and functions in MATLAB R2018B (MathWorks), along with functions from the EEGLAB (http://sccn.ucsd.edu/eeglab/) and the Chronux v2.12 (http://chronux.org/) toolboxes.

## LFP analysis

For the four animals with LFP recorded at 24,414 Hz, raw LFP signals were decimated channel by channel with an 8th order Chebyshev Type I low pass filter to a tenth of the original signal (2414 Hz). LFP for all animals was then pre-processed with the following steps: z-scoring the entire recording session, channel by channel; artifact rejection (manually removing noisy/broken channels, identifying trials with motion artifact on the majority of channels); common-mode referencing using the median signal (the median signal across all non-noisy channels in a region was calculated at every timepoint and subtracted from each channel to decrease common noise and minimize volume conduction). Common-mode referencing was performed independently for channels in each region, M1 and DLS.

We filtered the LFP signals to isolate and display the low-frequency (3–6 Hz) component of the signal. Filtering was performed using the EEGLAB function 'eegfilt.' To examine power across multiple frequency bands, we calculated movement-related LFP spectrograms and power spectra within each region using wavelets with the EEGLab function 'newtimef.' This function was also used to calculate the intertrial coherence (ITC) of LFP signals. We subsequently performed specific frequency band coherence analyses with the EEGLab function "cohgramc".

## Spiking analysis

*Unit modulation.* All spiking analyses were aligned to first reach onset (RO). To determine unit modulation (Fig. 2), peri-event time histograms (PETHs) were generated by averaging spiking activity for each neuron across trials in a session, locked to first reach onset, and binned at 50 ms (Fig. 2a, b, top). PETHs were then fit with a smoothing spline using a custom MATLAB function (Fig. 2a, b, bottom. To determine task-related unit modulation, we z-scored each unit's average firing rate for the session, found all peak prominences and times of activity from 2 s before reach onset to 2 s after reach onset with the MATLAB function "findpeaks", and identified the maximum peak of z-scored activity from −0.5 s to 0.5 s around first reach onset (Fig. 2f, g). This process was conducted separately for trials with first reach to pellet location A and trials with first reach to pellet location B across sessions.

*Determination of session type.* Sessions were categorized into the following based on task parameters, behavior curves, and consistency of unit spiking activity: (1) *Baseline* (BL), pellet at location A, reaching to A, consistent reach-modulated spiking activity; (2) *Automatic* (auto), pellet at location B, reaching mostly to A, consistent reach-modulated spiking activity; (3) *Variable* (var), pellet at location B, reaching to A and B, local maximum of minimum Fano factors per unit, aggregated, with significant deviation from baseline minimum Fano factors; (4) *Relearned* (rel), pellet at location B, reaching mostly to B, consistent reach-modulated spiking activity (Fig. 2c–e).

Fano factors describe how variable single-unit spiking activity is at a given timepoint across trials;[30,31] with increased spike count consistency within a given

time bin, there is a decrease in the Fano factor value for that time bin. To calculate Fano factors for each unit the following process was followed (Fig. 2c). First, for each animal, the minimum trial number across sessions was identified for sub-sampling. For each unit and each trial in a session, spike counts from −2 s to 2 s around RO were binned at 50 ms. Trials in a session were then sampled to the minimum trial number, and the Fano factor of each time bin was calculated, where Fano_timebin = standard deviation of spike counts in that time bin across trials, squared, divided by the mean spike count across trials for that time bin. For each unit in a given session, this process was repeated with 100 total trial sub-samples, and the final Fano factor per time bin was the median value of the 100 sub-sampled Fano factors per bin, with each unit having 80 total Fano factors spanning −2 s:RO:2 s per session.

To examine consistency of unit spiking activity, we identified the 5 Fano factors per unit per session with the lowest value (Fano$_{min5}$), thus allowing for timing-agnostic sampling of minimum variation in unit firing (Fig. 2d), as individual units consistently fired at different "preferred" timings relative to reach onset. To determine which sessions had a significantly different distribution of Fano$_{min5}$ from baseline, we used the MATLAB function 'kstest2'. Sessions which did not have units from both M1 and DLS were subsequently excluded from neural analyses. From remaining sessions, the *automatic* session was the session recorded closest in time before the *variable* session, and the *relearned* session the closest in time after for each animal. Since each animal followed unique curves, one animal did not have a *variable* or *relearned* session, one animal did not have a *relearned* session, and two animals did not have *automatic* sessions (Fig. 2e).

*Average task firing rate.* To calculate average task firing rate (Supplementary Fig. 2a, b), the minimum number of neurons in each region for an animal across sessions was identified. For A or B trials in a session, average firing rate for −2 s to 2 s around first reaches in the session across the sub-sampled neurons was calculated 1000 total times, with re-sampling of neurons each time to account for unit-to-unit firing rate variability. The median value of average firing rate was taken for each session per rat.

*Reach-related firing rate unit modulation.* To calculate reach-related spiking activity versus that at non-reach periods for the variable session, average trial firing rate from −0.5 s to 0.5 s around first reach onset was compared to the average session firing rate for that unit outside of the reaching time periods (Supplementary Fig. 2c).

*Template matching.* To assess how temporally consistent single-trial spiking activity was across session types, for each animal on each session we separated out trials with first reach to A and first reach to B. If there were at least 5 trials of one type for a given rat and session, each trial's spiking activity for a region (e.g., M1 units) was compared to the average template spiking from the remaining trials of that type[15]. Specifically, regional spiking activity from 500 ms before first reach RO to 500 ms after first reach RO was binned at 20 ms, smoothed with a 60 ms Gaussian kernel standard deviation, and concatenated across units for a given trial. Given the variable number of units in a session, the minimum number of units an animal had in a region, across sessions, was determined to be the number of units to subsample for this analysis; a minimum of 5 units was necessary for a session to be included. Each trial spiking activity, with sub-sampled units, was correlated to the average spiking activity from trials with the same sub-sampled units (Fig. 3a, d). This process was repeated 1000 times per set of trials, keeping the mean correlation across trials for each iteration; the median of these means is reported as the session trial-template correlation (Fig. 3c, e).

*Determination of directionally-tuned units.* To determine whether a given unit in a session was significantly tuned in directionality for reach to Location A or Location B, we fit a logistic regression model of spike count 200 ms around reach onset by reach location (e.g., short, A, B), using the MATLAB function "mnrfit". If the spike count at A reach onset was significantly higher than at B reach onset, a given unit was deemed "tuned to Location A" and vice versa.

*Dynamical system modeling.* We evaluated whether M1 spiking evolved according to a linear dynamical system by fitting an autoregressive model of M1 activity using ordinary linear regression. Specifically, for each unit, within-session trial spiking was binned at 20 ms, smoothed with a 200 point 5 ms Gaussian kernel, z-scored, and then trial-averaged. Subsequently, a dynamics model of the form $x_{t+1} = \mathbf{A}x_t$ was fit, where $x_t$ represented M1 data at time $t$. This process was repeated for each session type across sessions; A matrices and $R^2$ measures of model fit were recorded. All eigenvalues had positive and significant decay times—no exploding eigenvalues were observed and at least one eigenmode had a decay time greater than the bin size (20 ms) within each session for each rat.

*Population modulation.* To characterize population spiking activity modulation (Fig. 4a, b), z-scored unit activity from −2 s to 2 s around first RO for each trial was smoothed using a 5-point moving average, summed and then divided by the number of neurons for normalization. Trial activity for each session was then grouped into trials with first reach to A or trials with first reach to B. Normalized

population single-trial spiking activity was then averaged across trials to A or B within a session for both M1 and DLS units separately. All peak prominences and times of activity from −2 s to 2s around first RO were detected with the MATLAB function "findpeaks" with no minimum peak prominence. Maximum modulation prominence out of all local maxima in period from −2s to 2 s around first RO was identified as peak activity and timing of population reach-modulation (Fig. 4c–f, Supplementary Fig. 2d–g). Correlation of population peak activity (Fig. 4g) was computed by comparing the normalized M1 and DLS peak prominences from each trial of the *variable* session, across animals.

*Cross-area neural subspace.* Shared cross-area subspaces between M1 and DLS were identified using canonical correlation analysis (CCA), which defines axes that maximally correlates activity between the two areas[26,35]. Neural data in M1 and DLS were binned at 50 ms, and data from −2 s to 0.5 s surrounding first reach onset were concatenated across trials with first reach to A and trials with first reach to B separately. CCA models were then fit using the MATLAB function 'canoncorr', which involves transforming the data to have zero mean and unit standard deviation prior to computing canonical variables. The number of canonical variables (CVs) output by CCA is the minimum number of neurons in either M1 or DLS for that session. The $R^2$ value for each CV was computed using tenfold cross-validation. We randomly partitioned the full dataset into 10 folds and cycled through each fold, assigning one fold to be test data and the other nine to be the training data. We fit a CCA model to the training data, then project the test data onto this model, and compute the $R^2$ between the M1 and DLS projections onto the given CV. 95% significance was determined by comparison to a bootstrap distribution of top CV $R^2$ created from trial-shuffled data ($10^4$ shuffles), as described previously[26] (Fig. 5a, b). Only sessions with minimum 5 units in both areas were included for analysis; sessions with no significant CVs were subsequently removed from analyses. 17 sessions across 5 animals met initial criteria for CCA analysis inclusion for reaches to A—2 sessions had no significant CVs (11.8%) and were subsequently removed, 10 had 1 significant CV (58.8%), 4 had 2 significant CVs (23.5%), and 1 had 3 significant CVs (5.88%). 16 sessions across 5 animals with reaches to B met initial CCA analysis inclusion criteria—3 sessions had no significant CVs (18.8%) and were subsequently removed, 9 had 1 significant CV (56.3%), 3 had 2 significant CVs (18.8%), and 1 had 3 significant CVs (6.25%). For evaluating cross-area signals (Fig. 5c), only the top CV was used for consistency across datasets. To examine reaching-epoch modulation, we built similar models with data from −0.5 s to 0.5 s surrounding reach onset.

*Cross-area task representation.* To calculate the difference in cross-area activity before first reach versus during first reach (i.e., the relative modulation index), we defined a pre-reach period as -1 to -0.5 s before first reach onset, and reach period as −0.1 to 0.4 s around first reach onset (Fig. 5d). Cross-area median activity within a trial was calculated for each time period, and compared across session types (Fig. 5e–g). In those trials where the RMI is negative, the projected trial spiking on CV1 is more correlated during the pre-reach period than during the reach period. For time-limited CCA models built on reach-epoch specific activity, we identified the maximum subspace activity (i.e., peak trial subspace modulation) at −0.2 s to 0.2 s around reach onset (Supplementary Fig. 7b, c).

*Comparison of functional ICMS paradigms.* Three animals were used to probe differences in motor kinematics with patterned versus variable stimulation in M1. For the stimulation procedure, animals were initially anesthetized with a mixture of ketamine hydrochloride (100 mg/kg) and xylazine (16.67 mg/kg) delivered intraperitoneally. Supplementary 0.5–1 mL doses of the mixture were provided as needed, based on toe-pinch response. 32-channel tungsten microwire arrays (Tucker-Davis Technologies, ~50 kΩ input impedance at 1000 Hz) were implanted in M1 at a depth of 1500 μm, targeting cortical layer V.

For the patterned stimulation, as with prior studies[26,39], triplet biphasic trains of 200 μs per phase (100 μs inter-phase interval, 333 Hz triplet, 100–150 μA amplitude) were delivered at each electrode using a constant current stimulator (IZ2, TDT) controlled by a custom Synapse program (TDT). Stimulation was delivered for 2 s per session, for a total of 60 pulses per session. For the random pulse stimulation, pulse times were randomized such that there were 30 randomly timed pulses per second, repeated for a total of 2 s of stimulation; the same random pattern of stimulation was repeated across sessions. For both of these stimulation paradigms, there were often multiple channels across the array producing forelimb movement with stimulation.

Animals were placed in a prone position such that the contralateral forelimb remained free. Stimulation was initially delivered across the array, and channels that produced isolated forelimb movement were selected for further comparison. Within each animal, multiple sessions of either patterned burst stimulation or random pulse stimulation were delivered, differing only in temporal consistency or variability of the pulse pattern. Forelimb movement was video recorded at 20 frames per second, and trajectories of digit 3, digit 4, and the center of the paw were analyzed using DeepLabCut. Across all sessions within an animal, both patterned burst and random pulse, a 300 ms sliding window was simultaneously applied; windows with a comparable number of pulses (±1 pulse) were designated "trials"; multiple trials were identified within and across sessions. Trial identification was limited to the first 1 s within a session, due to limited forelimb movement in the

second half of the session, likely due to saturation. Subsequently, endpoint locations for each trial were defined as the x- and y-coordinates at the end of the 300 ms window relative to spatial location at the start of the trial window.

*Statistics.* Linear mixed effects models, with animal modeled as random effect on intercept and fixed effect for session type, were used to test the significance of differences across both behavioral and neural measures when comparing differences in group means. These models account for units or trials coming from the same animal, which are more correlated than those from different animals, thus providing a stricter computation of statistical significance. $P$ values were only reported for differences that were below the Bonferroni-corrected $p$ value for multiple comparisons ($\alpha_{BC} = 0.05$/(number of session type comparisons). For comparison of distribution broadness (Fig. 4f, Supplementary Fig. 2g), the Bartlett's test was used to determine whether samples came from populations with equal variances. Finally, we used the two-sample Kolmogorov-Smirnov test to test whether the underlying probability distributions of endpoint location with patterned burst or random pulse stimulation differed (Fig. 6e).

**Reporting summary.** Further information on research design is available in the Nature Research Reporting Summary linked to this article.

## Data availability
The data used for analyses supporting the findings of this study are available from the corresponding author upon reasonable request. Source data are provided with this paper.

## Code availability
The code used for analyses supporting the findings of this study are available from the corresponding author upon reasonable request, as the algorithms used were previously published.

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

## Acknowledgements

This work was supported by the Department of Veterans Affairs, Veterans Health Administration (to K.G., I01 RX001640), the National Institute of Neurological Disorders and Stroke (to S.K., F31 NS117010-01A1 and to D.D., F31 NS127514-01), the American Heart & Stroke Association (to S.M.L, pre-doctoral fellowship 17PRE33410530/2016), the UCSF Weill Neurohub (to K.G.) and the Northern California Institute for Research and Education (to K.G.). S.K. and D.D. were also supported by fellowship award from the UCSF Medical Scientist Training Program (2T32GM007618-39).

## Author contributions

S.K., S.L., K.G. designed the experiments. S.K. and S.L. carried out the experiments. S.K. primarily analyzed the data. D.D. analyzed the data from a neural dynamics perspective. L.G. assisted with the experimental setup and helped with analysis. P.K. assisted with data analysis and contributed code. S.K. wrote the paper; S.K. and K.G. revised together. All authors reviewed and edited the paper. K.G. supervised all aspects of this study.

## Competing interests

The authors declare no competing interests.

**Additional information**

