## [Peer Review File · Nature Communications]

Transition from predictable to variable corticostriatal patterning during behavioral explorationREVIEWER COMMENTS

Reviewer #1 (Remarks to the Author):

Nature Communications: NCOMMS-21-18972-T

Title: Tandem increases in motor cortex and striatal ensemble variability support behavioral exploration

Author: Sravani Kondapavulur, Stefan M. Lemke, David Darevsky, Ling Guo, Preeya Khanna, and Karunesh Ganguly

Comments:

The manuscript by Kondapavulur et al. examined the neural firings and the variability in the M1 and the DLS, and analyzed the temporal changes in the firings and variability in the face of a change in target location. Their results indicate that M1 and DLS flexibly transitioned between two modes, reliable neural pattern generation for automatic/precise movements and variable neural patterning for behavioral exploration. Although the temporal change of the firing pattern to re-learn new target location is very interesting, several major points of the experiment should be clarified.

Materials and Methods

1. Individual rats have dominant arms to take the pellet. The authors need to describe the L/R side of electrode together with the side of arm to take the pellet.
2. The authors should add schema or picture showing the location of the electrodes in the M1 and the DLS.
3. Reach onset (RO). The authors should describe the timing and the definition of the reach onset in detail. RO is the timing of taking behavior, first touch, or taking the pellet? Since time-locked analysis is critically important, unclear onset timing may lose the typical pattern of the firings in Variable.
4. Figure 6. Although the results may be interesting, it is difficult to find the relation or the significance to support the authors current findings.

Minor points

Line 904 "0.5-1 cc" should be "0.5-1 ml."

Line 1100 Figure 1f. Please explain what is r_1 and r_2 .

Line 1104 Figure 1d. Please explain what indicates single black circle?

Line 1104 Figure 2f, g, h, and i. It is difficult to understand the mean or the definition of Y-axes.

Line 1177 Legend says "tip of digit 5 (D5, blue), tip of digit 4 (D4, purple)" but Figure 6 shows D4 and D3.

Line 1187 Supp. Fig 1d and 1e. Please explain "Normalized reaction time" and "Normalized R1 time" in the legend or main text.

Reviewer #2 (Remarks to the Author):

This paper provides an illuminating analyses of changes in neuronal assemblies in not one but two brain areas associated with success in developing skilled reaching movements.

There is a load of work included and each figure is consequently dense with information that is sometimes hard to follow without some experience in the mathematical tools employed.

However the paper itself is clearly written and has a convincing argument for the conclusion that these cortical and striatal neurons are deeply involved both in the effective skilled movements and in their modification during training to a different target.

Reviewer #3 (Remarks to the Author):

Kondapavulur et al provide a description of how activity in primary motor cortex (M1) and dorsolateral striatum (DLS) change as rats learn to reach to a different target in a single pellet reach-to-grasp task. The key result is that there is a transition period as rats begin to reach to the new target during which reach-related modulation in M1 and DLS is lost, and variability in neural activity increases, before returning to a state similar to “baseline” learning. This is an important addition to the literature on how acquired skills are adapted to changing conditions. I think the main interpretation that there are discrete “exploratory” and “automatic” modes for M1-DLS coordination during skilled reaching is probably correct. Nonetheless, I have several significant questions regarding the methodology and interpretation of results that could confound that interpretation.

MAJOR COMMENTS:

1. I think a more detailed characterization of the behavior would improve the interpretation of the physiology.
 - a. Before the switch, rats generally reach directly to position “A”. After the switch, what is classified as a reach to “A” is really a “non-B, non-low amplitude” reach. Thus, the post-switch variability in “A” reach endpoint may be much higher than for “B” reaches, which is constrained by the definition that more than half the paw must cover the pellet. This is partially quantified in Figure 1C for a single rat, but I think should be quantified across rats as a function of session type (auto, var, rel). Is it possible that the changes in M1/DLS activity are related to variability in reach endpoints (as opposed to variability in whether reaches were directed to the “A” or “B” target).
 - b. The authors assert that the rats are making “smooth, fast reaches” across all session types, and this is key to the argument that the changes in M1/DLS activity are not linked to reach kinematics, but rather to “exploration” vs “automatic” modes. The speed of reaches is preserved based on supplemental fig 1d, e. However, it’s not clear to me that the reaches are necessarily “smooth” based on these data. For example, rats could make “wobbly” reaches with similar durations, or there could be increased variability in reach trajectories despite preserved speed. There are many ways “smoothness” and trajectory variability could be quantified across rats separately from reaction time and reach duration (for example, Azim E, Jiang J, Alstermark B, Jessell TM. Skilled reaching relies on a V2a propriospinal internal copy circuit. *Nature* 2014; 508: 357-363.; Bova A et al. Precisely timed dopamine signals establish distinct kinematic representations of skilled movements. *Elife* 2020; 9:). Only a single example of reach trajectories is shown in supplementary figure 1c. If the claim is that reach kinematics are similar in “var” sessions despite changes in neural activity, there should be more detailed characterization of reach trajectories and endpoints. Specifically, variability in reach trajectory and reach endpoints should be examined as a possible correlate of changes in M1/DLS activity.
 - c. Another possibility is that the modulation of M1 activity is related to the act of grasping. For example, Hyland et al (Hyland BI, Seeger-Armbruster S, Smither RA, Parr-Brownlie LC. Altered Recruitment of Motor Cortex Neuronal Activity During the Grasping Phase of Skilled Reaching in a Chronic Rat Model of Unilateral Parkinsonism. *J Neurosci* 2019; 39: 9660-9672.) found strong M1 modulation at the time of grasping. Is it possible that after several days of a pellet not being present at the “A” position, the rat stopped grasping and began feeling for the pellet with an open paw? This could be quantified by tracking the digits, or at least reviewing the videos to determine if the grasping movement changed qualitatively during “var” sessions.
 - d. Learning curves (% success vs number of trials/sessions) for initial training to target A and relearning after the switch to target B should be shown. During initial training, rats learn to transport their paw to the pellet and grasp it sequentially (e.g., Lemke et al, 2019). Once the rats have learned to reach and grasp at target A, do they only have to learn to reach to target B, or do they need to relearn the grasp as well? My prediction would be that the learning curve is steeper for target B once A has already been acquired because the rats already know how to grasp once they figure out where the pellet is.
2. Related to point 1c, when does the rat make a grasping attempt with respect to changes in single unit activity? This could be checked by locking PETHs to the end of the reach rather than reach onset. If single unit modulation is more closely locked to grasping than reach onset, this would suggest that the changes in neural activity may be more linked to changes in grasping than “exploration” vs “automatic” modes.

3. Line 144: "...there was no change in 3-6 Hz M1-DLS LFP coherence between session types (data not shown)". This seems important enough to show and have statistics, since the claim is that there are specific physiologic changes related to variability in reach targets.
4. I don't think the ICMS studies contribute to the interpretation of the recording studies. Suppose reach trajectories had remained consistent during variably timed ICMS. How would that change the interpretation of the recording results? In fact, it's hard for me to conceive of a model in which variably timed stimulation doesn't lead to variability in movement endpoints. I'm willing to be convinced that the outcomes of variable stimulation timing add new or confirmatory information to the neural recordings, but right now I don't see it.
5. I think the terms "transfer learning" and "motor adaptation" are ambiguous (line 619). At least, I cannot find clear, consistent definitions for them. I think of "transfer learning" as the acquisition of one task making it easier (or harder) to learn another, while I think of adaptation as learning new parameters for a modified version of a previously acquired task. This task seems more like an "adaptation" than "transfer learning" since this is basically the same task (maybe transfer learning would be learning a pasta-grasping task after a single pellet task?). I think that the authors are arguing that because rats were trained only on target A, the introduction of a second target makes this more like learning a new task than adapting the parameters of an old one (if they had been trained on multiple targets, then a novel target was introduced, would that be adaptation?). The evidence that this is "transfer learning" is that the neural dynamics associated with reaches to target A disappear in "var" sessions. I'm not sure if that follows without having shown that dynamics are preserved in tasks that would more clearly be classified as "adaptation" as opposed to "transfer learning". Maybe this is just semantics. In any case, if the authors wish to make a distinction between transfer learning and adaptation, they should define what they mean by each term and clarify why they believe their observations are more consistent with "transfer learning" than "adaptation".

Minor points/clarifications:

1. Line 99 "...after reaching to location A was > 50%." Does this mean the success rate for reaches to A was > 50%, or that > 50% of all reaches were targeted to location A?
2. Were the central and lateral locations the same across rats (corrected for paw preference)? How were central and lateral locations defined? In Figure 1, it looks like the A and B locations are both "lateral" – is this an artifact of the illustration or were the A and B sites symmetric around the midline of the reaching slot?
3. Did success rate systematically differ between central and lateral locations, regardless of which location was assigned as "A" or "B"? A switch from the "hard" to the "easy" location may have different characteristics electrophysiologically and behaviorally than a switch from the "easy" to the "hard" location. Also, it would just be interesting to know if one location is more difficult than the other.
4. How far apart are the "A" and "B" locations?
5. Are there ever trials where a pellet isn't delivered (intentionally or unintentionally)? Variable reward schedules cause more habit-like behavior. If there is a significant proportion of trials during training in which a pellet is not present (and especially if this fraction differs across rats), this could explain why rats take different amounts of time to switch to reaching to target B. That is, I would expect rats that took longer to switch to the B target to have had a larger fraction of absent pellet trials during training.
6. Lines 237-239. I had to read this sentence several times to understand it - "In M1, when comparing the first reach to Location A, spiking to the average neural template for first reaches to Location A, there was a drop in neural pattern consistency during the variable session as compared to the other sessions." I think it means that the pattern of spiking activity in M1 was less consistent during "var" sessions than BL, auto, or rel sessions.
7. Lines 258-259: "but that there's a consistent temporal ordering of unit modulation soon after,..." I could not follow what the authors were trying to convey here. It sounds to me like they're suggesting that individual units fire in the same order before and after the "var" session, but I don't think that's what they mean. I think they mean that there is a similar pattern of consistent modulation (BL, auto), then variability (var), then consistent modulation (rel) after the target switch, but I'm not sure.
8. Line 310 – "Across all animals is shown in Fig 4f." Not clear what is shown in figure 4f from this text. Presumably how the timing of peaks changed between session types?

9. Lines 567-570 – “Thus, this “unlearning” process of switching from previously rewarded ensemble activity to variable firing patterns likely involves a global network state shift, within both cognitive and motor circuits, towards generation of newer ensemble activity for reward.” I wouldn’t use the term “unlearning”, which to me suggests that the task has disappeared from memory (maybe it has, but my guess is rats would quickly be successful if the pellet were moved back to target A). Perhaps “adapting”?

10. Figure 2d – would make the vertical scale the same for both plots to facilitate comparison.

11. In the Figure 2a caption, it is implied that the same unit was recorded across sessions. This may be true, but no analyses are presented to support this assertion, other than that the units were recorded on the same electrode. Of course, it is impossible to be certain that the same unit was recorded, but there should be some indication that they are in fact the same unit if that is the claim. That said, I don’t think it’s critical to know whether the same units were recorded across days, so I think it would be fine to just change the caption so it’s clear that there was not an attempt to verify the unit’s identity across days. Fraser and Schwartz (Fraser GW, Schwartz AB. Recording from the same neurons chronically in motor cortex. *J Neurophysiol* 2012; 107: 1970-1978) described a fairly straightforward algorithm to test single unit identity across recording sessions, but again I don’t think this is necessary.

12. It may be worth commenting on how rats identify pellet location. There is evidence that they primarily use olfaction, and may not rely on vision at all (Whishaw IQ, Tomie JA. Olfaction directs skilled forelimb reaching in the rat. *Behav Brain Res* 1989; 32: 11-21.). I don’t think this significantly changes the authors’ interpretation that M1/DLS variability reflects an “exploratory” state (in fact, probably enhances it). However, it may be helpful for readers not familiar with rodent skilled reaching.

Reviewer #4 (Remarks to the Author):

This reviewer has been asked to comment specifically about the CCA analyses. The authors use CCA to find that activity shared between M1 and DLS appears to be more task-related in the automatic and relearned sessions, compared to variable sessions. This finding fits in well with the overall storyline of figuring out what is happening during the variable sessions. This is a reasonable use of CCA, although there are several points of interpretation that are currently unclear:

1) The authors consider only the top canonical variable (CV). To find the top CV, CCA was fit to the trial window (-2s to +0.5s, line 883), which includes the pre-reach period (-1 to -0.5s), the reach period (-0.1 to +0.4s), and other trial epochs. How does this choice affect interpretation? One could imagine, for example, that the M1-DLS “communication subspace” varies (e.g., rotates) considerably across these trial epochs, spanning multiple dimensions of population activity space in either M1 or DLS. How then, can we be confident that, as a 1D summary over multiple trial epochs, the top CV is indeed a good representation of M1-DLS interactions during both the pre-reach and reach trial epochs?

2) Related to the point 1), is it possible that task-related activity is present during the variable state, but it shows up in CV2 or CV3 rather than CV1?

3) Line 363 and 891: The authors state that “most” sessions included significant canonical variables between M1 and DLS, i.e., significant cross-area correlation. The authors should state how many sessions in which they could not identify any significant cross-area correlation (if it’s 49% of sessions, for example, that’s actually a lot of non-interactions).

4) In Fig 5d, the scatter of neural activity looks similar for Automatic (A reaches) and Relearned (A reaches). One might postulate that the neural activity should be quite different for Automatic (which is pre-learning) and Relearned (which is post-learning). Why is it reasonable that the activity looks similar in the two cases? What might that indicate about how the interactions between the two areas support (or not support) learning? (Same question for B reaches.)

5) Fig 5e: Is the sign of RMI arbitrary? It seems so because the sign of CV1 for M1 and DLS can both be flipped together to yield an equivalent CCA model. If that is true, does that affect the interpretation

of the results in Fig 5f and 5g, which show some positive and some negative RMI?

Minor comments:

- Line 76 and 347: If the authors are going to use the phrase "communication subspace," they should probably cite Semedo et al., Neuron 2019. They should also be careful about the use of this phrase (as in the Discussion, starting at Line 577), since a communication subspace refers to more than just the canonical variables returned by CCA, but also their relationship to the other activity patterns (if any) in M1 and DLS (i.e., are there activity patterns within each area that are not captured by the canonical variables?).

- Line 882: References 10-12 are cited for CCA. This might be a typo.

- Line 885: "mean activity in each group was subtracted." The authors should justify why this is needed and how it affects interpretation.

- Supp Fig 3B: Explain what the reader is supposed to take away from these CCA R^2 values. Also, it's unclear how they support the title of this figure: "Majority of M1 and DLS spiking not related to reach direction."

INTRODUCTION TO REVISED MANUSCRIPT

We thank the reviewers (R1/R2/R3/R4) for their thorough review and comments, which have significantly strengthened our approach. We also greatly appreciate the general enthusiasm for our study! We outline specific changes in response to the comments below:

Detailed response to Reviewer #1:

“The manuscript by Kondapavulur et al. examined the neural firings and the variability in the M1 and the DLS, and analyzed the temporal changes in the firings and variability in the face of a change in target location. Their results indicate that M1 and DLS flexibly transited between two modes, reliable neural pattern generation for automatic/precise movements and variable neural patterning for behavioral exploration. Although the temporal change of the firing pattern to re-learn new target location is very interesting, several major points of the experiment should be clarified.”

We thank for reviewer for their positive comments and appreciate the feedback.

“Materials and Methods

1. Individual rats have dominant arms to take the pellet. The authors need to describe the L/R side of electrode together with the side of arm to take the pellet.”

We appreciate that this was unclear in the methods, and we have clarified that the dominant reaching arm was identified prior to electrode implantation to determine implantation hemisphere, particularly in Methods, Surgery (**lines 715-716**) and Methods, Behavioral training (**lines 750-756**).

“2. The authors should add schema or picture showing the location of the electrodes in the M1 and the DLS.”

We thank the reviewer for providing this clarifying suggestion and have added a schema of M1 and DLS locations in the Figure 1a inset. The figure legend has been updated accordingly (**line 1166**).

“3. Reach onset (RO). The authors should describe the timing and the definition of the reach onset in detail. RO is the timing of taking behavior, first touch, or taking the pellet? Since time-locked analysis is critically important, unclear onset timing may lose the typical pattern of the firings in Variable. “

We have included the definition of reach onset (RO) in the Methods section, i.e., initiation of forward displacement of the paw after the paw has completely rotated from flexion to extension, in Methods, Behavioral analysis (**lines 779-781**).

“4. Figure 6. Although the results may be interesting, it is difficult to find the relation or the significance to support the authors current findings.”

We thank Reviewer 1 (and Reviewer 3) for pointing out that the findings in Figure 6 are difficult to connect back to the main findings. We have added additional text to better frame our approach and to motivate these experiments (**lines 440-450**). In brief, there is now clear evidence that reliable and predictable patterns of neural activity are associated with robust movement control (e.g., Churchland and Shenoy, Nature, 2012 and JA Gallego, et al, Neural Manifolds for the Control of Movement, Neuron 2017). Notably, it is quite likely that our observed predictable neural sequences of firing in M1/DLS are analogous to these findings (indeed we also quantify the predictable dynamics).

The main question, then, is whether a loss of reliable sequencing and variable firing is still capable of driving movements? While the ICMS results are certainly not in awake animals, they do suggest that variable firing patterns in M1 are still capable of driving movements, albeit with end-point variability. While we are certainly open to removing these results, we respectfully suggest that they provide support for the notion that reliable and variable patterning in M1 are still potentially movement potent. These also motivate future, and more challenging work, in which more naturalistic patterning of M1 (e.g., with emerging methods such as holographic stimulation) might also be testable.

Minor points

Line 904 "0.5-1 cc" should be "0.5-1 ml."

We have updated this line to "0.5-1mL."

Line 1100 Figure 1f. Please explain what is r1 and r2.

We have updated the legend for Figure 1f to reflect that r1 and r2 are the first and second reach onsets, respectively, within the demonstrated trial (**lines 1172-1174**).

Line 1104 Figure 1d. Please explain what indicates single black circle?

We believe that this question is regarding Figure 1c. The legend for Figure 1c explains that the single black circle indicates pellet location.

Line 1104 Figure 2f, g, h, and i. It is difficult to understand the mean or the definition of Y-axes.

We have updated the figure legend to clarify the metrics in Figure 2f-i (**lines 1188-1193**)

Line 1177 Legend says "tip of digit 5 (D5, blue), tip of digit 4 (D4, purple)" but Figure 6 shows D4 and D3.

We have corrected the legend to accurately reflect D4 and D3 as depicted in the figure (**line 1252**).

Line 1187 Supp. Fig 1d and 1e. Please explain "Normalized reaction time" and "Normalized R1 time" in the legend or main text.

We have updated the legend for Supplemental Figure 1 to reflect the process of normalization, which involved subtracting the minimum value across sessions (1 minimum value per rat) for the reaction time (Supp. Fig 1g) and the first reach time (Supp. Fig 1h) separately (**lines 1270-1272,1277-1278**).

Detailed response to Reviewer #2:

“This paper provides an illuminating analyses of changes in neuronal assemblies in not one but two brain areas associated with success in developing skilled reaching movements. There is a load of work included and each figure is consequently dense with information that is sometimes hard to follow without some experience in the mathematical tools employed. However the paper itself is clearly written and has a convincing argument for the conclusion that these cortical and striatal neurons are deeply involved both in the effective skilled movements and in their modification during training to a different target.”

We sincerely thank the reviewer for their kind summary regarding the impact of the project described.

Detailed response to Reviewer #3:

“Kondapavulur et al provide a description of how activity in primary motor cortex (M1) and dorsolateral striatum (DLS) change as rats learn to reach to a different target in a single pellet reach-to-grasp task. The key result is that there is a transition period as rats begin to reach to the new target during which reach-related modulation in M1 and DLS is lost, and variability in neural activity increases, before returning to a state similar to “baseline” learning. This is an important addition to the literature on how acquired skills are adapted to changing conditions. I think the main interpretation that there are discrete “exploratory” and “automatic” modes for M1-DLS coordination during skilled reaching is probably correct. Nonetheless, I have several significant questions regarding the methodology and interpretation of results that could confound that interpretation.”

We thank for reviewer for their positive reviews regarding the impact of this study and appreciate the feedback.

“MAJOR COMMENTS:

1a. I think a more detailed characterization of the behavior would improve the interpretation of the physiology. Before the switch, rats generally reach directly to position “A”. After the switch, what is classified as a reach to “A” is really a “non-B, non-low amplitude” reach. Thus, the post-switch variability in “A” reach endpoint may be much higher than for “B” reaches, which is constrained by the definition that more than half the paw must cover the pellet. This is partially quantified in Figure 1C for a single rat, but I think should be quantified across rats as a function of session type (auto, var, rel). Is it possible that the changes in M1/DLS activity are related to variability in reach endpoints (as opposed to variability in whether reaches were directed to the “A” or “B” target).”

We thank the reviewer for this observation and introduction of alternate behavioral interpretations. For animals with top-down videos, we quantified spread via standard deviation in the X and Y directions of the camera image, as well as collapsed average standard deviation, categorized by session type. Average spread was greatest in the baseline and automatic conditions. In the relearned condition, there was significantly less deviation in the Y and averaged X-Y directions than there was at baseline; no other conditions (automatic, variable) had significantly different deviations than baseline. We have added this data into Supplementary Figure 2 (**lines 1280-1285**), and have shifted the remaining supplementary figures accordingly. We interpret this to mean that variability in end-points are relatively stable as the mean target goal is shifting during exploration, as detailed in Results, Loss of consistent reach-locked M1 and DLS neural spiking during exploration (**lines 209-216**).

“1b. The authors assert that the rats are making “smooth, fast reaches” across all session types, and this is key to the argument that the changes in M1/DLS activity are not linked to reach kinematics, but rather to “exploration” vs “automatic” modes. The speed of reaches is preserved based on supplemental fig 1d, e. However, it’s not clear to me that the reaches are necessarily “smooth” based on these data. For example, rats could make “wobbly” reaches with similar durations, or there could be increased variability in reach trajectories despite preserved speed. There are many ways “smoothness” and trajectory variability could be quantified across rats separately from reaction time and reach duration (for example, Azim E, Jiang J, Alstermark B, Jessell TM. Skilled reaching relies on a V2a propriospinal internal copy circuit. Nature 2014; 508: 357-363.; Bova A et al. Precisely timed dopamine signals establish distinct kinematic representations of skilled movements. Elife 2020; 9:). Only a single example of reach trajectories is shown in supplementary figure 1c. If the claim is that reach kinematics are similar in “var” sessions despite changes in neural activity, there should be more detailed characterization of reach trajectories and endpoints. Specifically, variability in reach trajectory and reach endpoints should be examined as a possible correlate of changes in M1/DLS activity.”

We appreciate the reviewer’s comment asking for better characterization of reaching kinematics. In E Azim, et al, Nature 2014, the authors quantified velocity over distance, and they statistically compared “direction reversals” during reaching, which we qualitatively do not see visually in this experimental paradigm. In A Bova, et al, ELife 2020, the comparable metric for our study would be maximum reach velocity. Thus, we have added the following to Supplemental Figure 1: 1) panel ‘i’ demonstrating velocity profiles locked to grasp across session types for an example animal, additionally demonstrating that there are no “direction reversals” on average after switching location, and 2) panel ‘j’ demonstrating that maximum velocity during reach-to-grasp does not change across session types (for all animals) (**lines 1272-1275**). These findings were also added to Results, Loss of consistent reach-locked M1 and DLS neural spiking during exploration (**lines 146-148**). With regard to better characterization of reach endpoints, we have quantified the spread of reach endpoints across session types, detailed in the response to point 1a.

“1c. Another possibility is that the modulation of M1 activity is related to the act of grasping. For example, Hyland et al (Hyland BI, Seeger-Armbruster S, Smither RA, Parr-Brownlie LC. Altered Recruitment of Motor Cortex Neuronal Activity During the Grasping Phase of Skilled Reaching in a Chronic Rat Model of Unilateral Parkinsonism. J Neurosci 2019; 39: 9660-9672.) found strong M1 modulation at the time of grasping. Is it possible that after several days of a pellet not being present at the “A” position, the rat stopped grasping and began feeling for the pellet with an open paw? This could be quantified by tracking the digits, or at least reviewing the videos to determine if the grasping movement changed qualitatively during “var” sessions. “
When we qualitatively reviewed the videos, there was no difference in grasping movements after switching the pellet location – that is the grasping was still quick, and most every reach was followed by a grasp. To further explore whether the variable sessions had more reaches and fewer grasps, we examined the proportion of reaches that had grasps by animal across session types. There was no significant difference in the proportion of reaches followed by grasping movements across session types, as seen in Supp Fig 1e (**lines 1268-1269**). Additionally, we have included a series of videos demonstrating reaching behavior across the session types, with qualitatively preserved reach-to-grasp sequences (Supp Vid 1, **lines 1326-1336**). Finally, we would also like to clarify that both M1 and DLS modulation are changing during this behavioral paradigm, not just M1. Canonically, DLS is involved non-dexterous forelimb movement (AK Dhawale, et. al, The basal ganglia control the detailed kinematics of learned motor skills, Nature Neuroscience 2021, SM Lemke, et. al, Emergent modular neural control drives coordinated motor actions, Nature Neuroscience 2019), and this patterning simultaneously changes alongside that of M1, indicating that this likely represents a change in reach-to-grasp combined strategy rather than grasping behavior alone.

“1d. Learning curves (% success vs number of trials/sessions) for initial training to target A and relearning after the switch to target B should be shown. During initial training, rats learn to transport their paw to the pellet and grasp it sequentially (e.g., Lemke et al, 2019). Once the rats have learned to reach and grasp at target A, do they only have to learn to reach to target B, or do they need to relearn the grasp as well? My prediction would be that the learning curve is steeper for target B once A has already been acquired because the rats already know how to grasp once they figure out where the pellet is.”

We have added these learning curves for our cohort as Supplementary Figure 1, panel A (top), in addition to the curve for relearning after switch to target B (panel A, bottom) (**lines 1264-1265**). To address the reviewer’s question, the rats only need to learn reach to target B, and quickly apply grasping knowledge as seen by the higher accuracy (>50%) by post-switch session 2 in animals that reached the variable state more quickly. Thus, as the reviewer predicted, there is a quicker learning curve for target B if they are able to reach the variable state.

“2. Related to point 1c, when does the rat make a grasping attempt with respect to changes in single unit activity? This could be checked by locking PETHs to the end of the reach rather than reach onset. If single unit modulation is more closely locked to grasping than reach onset, this would suggest that the changes in neural activity may be more linked to changes in grasping than “exploration” vs “automatic” modes.”

We took this advice and re-ran the analyses from Figure 2 locked to first grasp instead of to first reach. We found that there was a similarly significant increase in minimum Fano factors during the variable session across 4/5 animals. One animal did not demonstrate this increase, likely due to high baseline variability when aligned to first grasp. Using the same variable sessions as previously identified, we also calculated trial-averaged z-scored unit modulation in M1 and DLS during the period -0.5s : first grasp : 0.5s. In M1, there was a significant decrease in unit modulation from baseline and automatic to variable, and significant increase in unit modulation from variable to relearned. In DLS, there was a significant decrease in unit modulation from the automatic to variable sessions, and significant increase from variable to relearned. We have added these findings to the Results section and to Supplemental Figure 5. Please see Results, Loss of consistent grasp-locked M1 and DLS neural spiking during transfer learning (**lines 237-253**) and Supplemental Figure 5 (**lines 1305-1311**).

“3. Line 144: “...there was no change in 3-6 Hz M1-DLS LFP coherence between session types (data not shown)”. This seems important enough to show and have statistics, since the claim is that there are specific physiologic changes related to variability in reach targets.”

We appreciate the insight that this data would enrich the discussion regarding neurophysiologic correlates related to movement variability. We have included a Supplementary Figure 4 (**lines 1300-1304**) that reflects

the finding that baseline session reaches to A and variable session reaches to B have no significant difference in 3-6Hz M1-DLS LFP coherence and have updated the corresponding results section: Results, Stability of 3-6Hz M1-DLS coherence during variable state (**lines 226-235**).

“4. I don’t think the ICMS studies contribute to the interpretation of the recording studies. Suppose reach trajectories had remained consistent during variably timed ICMS. How would that change the interpretation of the recording results? In fact, it’s hard for me to conceive of a model in which variably timed stimulation doesn’t lead to variability in movement endpoints. I’m willing to be convinced that the outcomes of variable stimulation timing add new or confirmatory information to the neural recordings, but right now I don’t see it.”

We have added additional text to better frame our approach and to motivate these experiments (**lines 440-450**); please also see the response to R1, point 4. We agree that the likely outcome was that variable ICMS patterning in M1 (as compared to reliable patterning) leads to more variable movements. However, as the reviewer points out, it was still possible that this is not the case (especially as, to the best of our knowledge, this experiment has not been done). Such a result is particularly conceivable because there are multiple other downstream areas (e.g., the red nucleus, reticular nucleus and the spinal cord) which could easily filter the output of M1 to make movements more similar. We thus felt it important to demonstrate that M1 variability (albeit using ICMS) may in fact be directly mapped to end-point variability. In our view, when the ICMS results are considered together with our recording results, they really highlight the full potential of M1 as a controller, not just as a reliable pattern generator but also an active modulator of behavioral variability and exploration. One could also consider simply removing Figure 6. However, we would like to point out, that in that case, we are less directly able to claim that M1 patterning (consistent or variable) has the potential to be directly mapped to movement control. Finally, in our view, the ICMS results perhaps help motivate future experiments in awake behaving animals where M1 patterning can be modulated with emerging methods in order to assess how ensemble patterns may be directly mapped to goal-directed movement control.

“5. I think the terms “transfer learning” and “motor adaptation” are ambiguous (line 619). At least, I cannot find clear, consistent definitions for them. I think of “transfer learning” as the acquisition of one task making it easier (or harder) to learn another, while I think of adaptation as learning new parameters for a modified version of a previously acquired task. This task seems more like an “adaptation” than “transfer learning” since this is basically the same task (maybe transfer learning would be learning a pasta-grasping task after a single pellet task?). I think that the authors are arguing that because rats were trained only on target A, the introduction of a second target makes this more like learning a new task than adapting the parameters of an old one (if they had been trained on multiple targets, then a novel target was introduced, would that be adaptation?). The evidence that this is “transfer learning” is that the neural dynamics associated with reaches to target A disappear in “var” sessions. I’m not sure if that follows without having shown that dynamics are preserved in tasks that would more clearly be classified as “adaptation” as opposed to “transfer learning”. Maybe this is just semantics. In any case, if the authors wish to make a distinction between transfer learning and adaptation, they should define what they mean by each term and clarify why they believe their observations are more consistent with “transfer learning” than “adaptation”.”

We thank the reviewer for bringing up this point. First, we can begin with why this behavior is less consistent with typical notions of adaptation: adaptation is the recovery of motor performance within a changed environment, a process which occurs through error-based learning (R Shadmehr, FA Mussa-Ivaldi, J Neurosci 1994; RD Seidler, et. al, Adv Exp Med Biol 2013); moreover, most experimental paradigms include testing of an “aftereffect” following the period of adaptation. Given our paradigm is so different from such experiments, we did not want to create confusion by adopting the terminology of adaptation. We also do agree that this behavior is not transfer learning in the classical sense either, as detailed by the reviewer above. Thus, we have updated the entire text to describe the paradigm as behavioral exploration in response to errors, leading to “relearning” or convergence to a new strategy.

Minor points/clarifications:

1. Line 99 “...after reaching to location A was > 50%.” Does this mean the success rate for reaches to A was > 50%, or that > 50% of all reaches were targeted to location A?

We have clarified this sentence to reflect the former, that success rate for reaches to A was greater than 50% (**lines 98-99**).

2. Were the central and lateral locations the same across rats (corrected for paw preference)? How were central and lateral locations defined? In Figure 1, it looks like the A and B locations are both “lateral” – is this an artifact of the illustration or were the A and B sites symmetric around the midline of the reaching slot?

The central and lateral locations were not exactly the same across rats, as different rats had different “preferred” reaching locations with regard to cross-directionality and amplitude (i.e. a right paw-reaching rat could reach either straight centrally or across to the left without encountering the wall, with sufficient distance away from the wall such that reaching was encouraged over licking); this first location was defined as ‘A.’ ‘B’ was a location that was one pellet arm width (1cm) over from reach location A, either towards the center or laterally, with the same distance away from the center of the slit in the wall as A was (i.e. different angle from midline, same reach amplitude). We thank the reviewer for pointing out the ambiguity between locations A and B within the illustration. The A and B locations were not symmetric around midline, and this has now been updated in the illustration to be more accurate. We have additionally clarified A/B pellet within Methods, Behavioral training (**lines 750-756**).

3. Did success rate systematically differ between central and lateral locations, regardless of which location was assigned as “A” or “B”? A switch from the “hard” to the “easy” location may have different characteristics electrophysiologically and behaviorally than a switch from the “easy” to the “hard” location. Also, it would just be interesting to know if one location is more difficult than the other.

Success rate was similarly > 50% to A at baseline, whether the location was central or lateral. Additionally, to control for this possibility of differences in difficulty, both types of switches were represented (i.e. from central to lateral, and lateral central), as detailed in Results: Transfer learning of an automatic skill is a multi-day process (**line 100-102**).

4. How far apart are the “A” and “B” locations?

As now detailed in point 2, the distance between A and B was 1cm, with the same amplitude away from the wall.

5. Are there ever trials where a pellet isn’t delivered (intentionally or unintentionally)? Variable reward schedules cause more habit-like behavior. If there is a significant proportion of trials during training in which a pellet is not present (and especially if this fraction differs across rats), this could explain why rats take different amounts of time to switch to reaching to target B. That is, I would expect rats that took longer to switch to the B target to have had a larger fraction of absent pellet trials during training. (R3):

We thank the reviewer for bringing up this nuance. We calculated the proportion of trials during early learning where there erroneously was no reward presented and compared this to the number of days it took to reach the variable session. We have added this as Supplemental Figure 1d, demonstrating that among the 5 animals with a variable session, the relationship between no-reward-available trials and variable day has $R^2=0.743$, $p=0.0604$ (**lines 1267-1268**). It is possible that variable reward schedules could play some role in causing habit-like, automatic behavior, as does number of trials to A.

6. Lines 237-239. I had to read this sentence several times to understand it - “In M1, when comparing the first reach to Location A, spiking to the average neural template for first reaches to Location A, there was a drop in neural pattern consistency during the variable session as compared to the other sessions.” I think it means that the pattern of spiking activity in M1 was less consistent during “var” sessions than BL, auto, or rel sessions.

We have simplified this sentence to reflect that during the variable session, there was a drop in trial-template neural pattern consistency in M1 (**lines 267-269**).

7. Lines 258-259: “but that there’s a consistent temporal ordering of unit modulation soon after,…” I could not follow what the authors were trying to convey here. It sounds to me like they’re suggesting that individual units fire in the same order before and after the “var” session, but I don’t think that’s what they mean. I think they mean that there is a similar pattern of consistent modulation (BL, auto), then variability (var), then consistent modulation (rel) after the target switch, but I’m not sure.

We have edited this sentence to clarify that we mean the latter point, that there is a return to temporal consistency of firing across units at first reach onset after the variable session (**lines 287-289**).

8. Line 310 – “Across all animals is shown in Fig 4f.” Not clear what is shown in figure 4f from this text. Presumably how the timing of peaks changed between session types?

Figure 4f details the peak population firing rate for a trial, across trials, across animals, grouped by session type; this has been clarified in the text (**lines 1215-1216**).

9. Lines 567-570 – “Thus, this “unlearning” process of switching from previously rewarded ensemble activity to variable firing patterns likely involves a global network state shift, within both cognitive and motor circuits, towards generation of newer ensemble activity for reward.” I wouldn’t use the term “unlearning”, which to me suggests that the task has disappeared from memory (maybe it has, but my guess is rats would quickly be successful if the pellet were moved back to target A). Perhaps “adapting”?

We thank the author for pointing out this distinction and have updated the phrasing per the response more fully explained in Major Comment 5 above (**lines 620-623**).

10. Figure 2d – would make the vertical scale the same for both plots to facilitate comparison.

We thank the reviewer for this comment. We have decided to leave the vertical scale as is to highlight the point of the figure, that each animal has an increase in Fano factors of unit spiking during the variable session. Comparison across animals is less meaningful, due to the units being sampled having different baseline modulation from animal to animal.

11. In the Figure 2a caption, it is implied that the same unit was recorded across sessions. This may be true, but no analyses are presented to support this assertion, other than that the units were recorded on the same electrode. Of course, it is impossible to be certain that the same unit was recorded, but there should be some indication that they are in fact the same unit if that is the claim. That said, I don’t think it’s critical to know whether the same units were recorded across days, so I think it would be fine to just change the caption so it’s clear that there was not an attempt to verify the unit’s identity across days. Fraser and Schwartz (Fraser GW, Schwartz AB. Recording from the same neurons chronically in motor cortex. J Neurophysiol 2012; 107: 1970-1978) described a fairly straightforward algorithm to test single unit identity across recording sessions, but again I don’t think this is necessary.

We thank the reviewer for highlighting the ambiguity, and we are not claiming that the same unit was recorded across days. We have updated the figure legend accordingly (**lines 1182-1183, 1194**).

12. It may be worth commenting on how rats identify pellet location. There is evidence that they primarily use olfaction, and may not rely on vision at all (Whishaw IQ, Tomie JA. Olfaction directs skilled forelimb reaching in the rat. Behav Brain Res 1989; 32: 11-21.). I don’t think this significantly changes the authors’ interpretation that M1/DLS variability reflects an “exploratory” state (in fact, probably enhances it). However, it may be helpful for readers not familiar with rodent skilled reaching.

We agree with the evidence from Wishaw and Tomie, 1989, that rodents identify pellet location via olfaction, and with the suggestion that this would be helpful for the broader reading audience. Thus, we have included this information within Methods, Behavioral training (**lines 766-767**).

Detailed response to Reviewer #4:

This reviewer has been asked to comment specifically about the CCA analyses. The authors use CCA to find that activity shared between M1 and DLS appears to be more task-related in the automatic and relearned sessions, compared to variable sessions. This finding fits in well with the overall storyline of figuring out what is happening during the variable sessions. This is a reasonable use of CCA, although there are several points of interpretation that are currently unclear:

We thank the reviewer for the positive review on use of CCA and appreciate the feedback.

“1) The authors consider only the top canonical variable (CV). To find the top CV, CCA was fit to the trial window (-2s to +0.5s, line 883), which includes the pre-reach period (-1 to -0.5s), the reach period (-0.1 to +0.4s), and other trial epochs. How does this choice affect interpretation? One could imagine, for example, that the M1-DLS “communication subspace” varies (e.g., rotates) considerably across these trial epochs, spanning multiple dimensions of population activity space in either M1 or DLS. How then, can we be confident that, as a 1D summary over multiple trial epochs, the top CV is indeed a good representation of M1-DLS interactions during both the pre-reach and reach trial epochs?”

Our motivation to use the broad model was first driven by our past work. Use of the top CV determined from a broad time period for comparison across animals and time has been used across two different experimental paradigms, including early learning and stroke recovery (T Veuthey*, K Derosier*, et al, Nature Communications 2020; L Guo, et al, Cell Reports 2021). One of the findings from these papers was there was always some degree of modulation in the CCA subspace, i.e. both prior to and during reach. Learning appears to substantially increase subspace activation. A second reason to use the broad time period here is that we did not want to only fit a model during reach for the variable session, which would limit conclusions surrounding when M1-DLS interactions are most modulated. Based on the past results, we anticipated that the broader model is perhaps able to better capture temporal coordination, and we can examine the RMI as a measure of changes in *task-related* subspace activation for each of the session types.

Here, we also followed a similar approach. One of the findings of this approach for the “broad window” CCA analysis is that the reach epoch is when M1-DLS cross-area subspace activity is maximal during baseline and relearned states (i.e. Fig 5c). This allowed us to then calculate the RMI by comparing pre-reach to reach period activations of this subspace. We then utilized the RMI to determine how CCA subspace reach-related modulation changed during the variable session; we found that dominant CV subspace activity was less increased by reaching, as evidenced by decreased RMI.

We can also answer this question of broad-time period CCA model validity by comparing R^2 values of sub-models built on pre-reach (-1.5s:-0.5s relative to reach onset) and reach (-0.5:reach onset:0.5s) periods to that of the full model. For this analysis, only sessions with all three R^2 values were included for analysis. For models built on reach to A, there was no significant difference between R^2 for pre-reach (pre) vs broad ($p=0.677$), or reach vs. broad ($p=0.585$). This indicated that both epoch-based CCA models and broad time period-based CCA models are similarly generalizable.

Rat #	Session type	A/B	CV #	R2 (pre)	R2 (reach)	R2 (broad)
2	BL	A	1	0.141	0.116	0.182
3	Var	A	1	0.0443	0.0602	0.0717
3	Rel (5)	A	1	0.114	0.106	0.0564
4	Auto	A	1	0.0190	0.0048	0.0093
5	BL	A	1	0.102	0.0648	0.149
5	Var	A	1	0.112	0.113	0.131
5	Rel (3)	A	1	0.0856	0.0751	0.0602
6	Rel (3)	A	1	0.169	0.214	0.171

For models built on reach to B, there was significantly lower R^2 for pre vs broad ($p=0.0162$), with no significant difference between reach vs. broad ($p=0.102$).

3	Auto	B	1	0.0892	0.204	0.179
3	Var	B	1	0.0468	0.0612	0.0754
3	Rel (5)	B	1	0.0297	0.0889	0.116

5	Var	B	1	0.109	0.0602	0.0801
5	Rel (5)	B	1	0.239	0.0927	0.268
5	Rel (6)	B	1	0.348	0.114	0.343
6	Var	B	1	0.287	0.274	0.282
6	Rel (3)	B	1	0.0607	0.184	0.236
6	Rel (4)	B	1	0.0413	0.147	0.188
6	Rel (5)	B	1	0.0810	0.357	0.306
6	Rel (6)	B	1	0.177	0.210	0.218

These results are consistent with our past results, that there is some degree of cross-area transmission across time during the experiment; the extent of transmission can be modulated by learning and exploratory behaviors. Of note, many sessions only had one significant CV, and thus the top CV was used for similarity of comparison across animals and sessions. 17 sessions across 5 animals met criteria for CCA analysis inclusion for reaches to A, as detailed in the Methods section – 2 had no significant CVs (11.8%), 10 had 1 significant CV (58.8%), 4 had 2 significant CVs (23.5%), and 1 had 3 significant CVs (5.88%). 16 sessions across 5 animals with reaches to B met criteria – 3 had no significant CVs (18.8%), 9 had 1 significant CV (56.3%), 3 had 2 significant CVs (18.8%), and 1 had 3 significant CVs (6.25%).

Finally, to examine whether reach-epoch limited models might demonstrate preserved reach-related subspace activations in contrast with our findings in Fig. 5f,g, we compared trial peak subspace modulation at -0.2s to 0.2s around reach onset across session types. Strikingly, across M1 and DLS we also found lower subspace modulation for both A and B reaches during the variable session, as compared to the automatic state and relearned state, respectively. This data has been added as Supplemental Figure 7b,c (**lines 1321-1324**), and to Results, Changes in M1-DLS cross-area subspace modulation (**lines 421-430**), demonstrating that whether the broader time window or reach-related window is used to build the CCA model, there is a drop in reach-related cross-area activity.

“2) Related to the point 1), is it possible that task-related activity is present during the variable state, but it shows up in CV2 or CV3 rather than CV1?”

We thank the reviewer for pointing out this alternate possibility. For the valid variable sessions with reaches to A, only one session (from one rat) had a second significant CV. Because we only had a single session, we tested whether the distribution of pre-reach and reach-related activity was significantly different using the Kolmogorov-Smirnov test (K-S test) of two samples, and found no significant difference (M1: ks2stat = 0.0769, p = 0.411; DLS: ks2stat = 0.0538, p = 0.835).

For the valid variable sessions with reaches to B, we similarly performed K-S tests by session to determine whether there was significantly different activity during the reach period in non-top CVs:

Rat #	CV #	M1 ks2stat	M1 p-value	DLS ks2stat	DLS p-val
3	2	0.0839	0.0365	0.0500	0.476
4	(only 1 CV)	-	-	-	-
5	2	0.0359	0.960	0.0564	0.552
6	2	0.0236	0.998	0.0636	0.208
6	3	0.0382	0.810	0.0582	0.301

Thus, for the vast majority of variable sessions, even the CVs that were not the top CV did not contain reach-related activity as compared to prior to reach onset.

“3) Line 363 and 891: The authors state that “most” sessions included significant canonical variables between M1 and DLS, i.e., significant cross-area correlation. The authors should state how many sessions in which they could not identify any significant cross-area correlation (if it’s 49% of sessions, for example, that’s actually a lot of non-interactions).”

As detailed in the response to point 1, 17 sessions across 5 animals met criteria for CCA analysis inclusion for reaches to A, as detailed in the Methods section – 2 had no significant CVs (11.8%), 10 had 1 significant CV (58.8%), 4 had 2 significant CVs (23.5%), and 1 had 3 significant CVs (5.88%). 16 sessions across 5 animals with reaches to B met criteria – 3 had no significant CVs (18.8%), 9 had 1 significant CV (56.3%), 3 had 2

significant CVs (18.8%), and 1 had 3 significant CVs (6.25%). We have updated Methods, Cross-area neural subspace accordingly (**lines 950-955**).

“4) In Fig 5d, the scatter of neural activity looks similar for Automatic (A reaches) and Relearned (A reaches). One might postulate that the neural activity should be quite different for Automatic (which is pre-learning) and Relearned (which is post-learning). Why is it reasonable that the activity looks similar in the two cases? What might that indicate about how the interactions between the two areas support (or not support) learning? (Same question for B reaches.)”

What we’re seeing here might just be mass driving of M1 to DLS rather than the details of A vs. B; that is, M1-DLS communication during reaching is a consistent M1 to DLS drive in the Automatic and Relearned states. An alternate possibility is that for the Automatic state, there is motor noise (i.e. model for A and model for B are similar) such that reaches to B are within the motor noise of reaches to A, and for the Relearned state there could be two co-existing models. However, given the fact that overwhelming majority neurons aren’t selectively modulated for either reach type (Supp. Fig. 6a), evidence points towards the “mass driving” hypothesis.

“5) Fig 5e: Is the sign of RMI arbitrary? It seems so because the sign of CV1 for M1 and DLS can both be flipped together to yield an equivalent CCA model. If that is true, does that affect the interpretation of the results in Fig 5f and 5g, which show some positive and some negative RMI?”

The sign of CV1 is arbitrary - therefore we chose the sign of CV1 that enabled mean activity along CV1 within a session to be the same direction across sessions (e.g. positive). Thus, we could directly compare RMI of trials across sessions and across animals. In those trials where the RMI is negative, the projected trial spiking on CV1 is less correlated during reach than it is prior to reach.

Minor comments:

- Line 76 and 347: If the authors are going to use the phrase "communication subspace," they should probably cite Semedo et al., Neuron 2019. They should also be careful about the use of this phrase (as in the Discussion, starting at Line 577), since a communication subspace refers to more than just the canonical variables returned by CCA, but also their relationship to the other activity patterns (if any) in M1 and DLS (i.e., are there activity patterns within each area that are not captured by the canonical variables?).

We thank the reviewer for bringing up this oversight, and have cited Semedo, et. al. accordingly when introducing the concept. Given that we are only examining the CCA subspace and not the relationship to other local activity patterns, we have corrected references to “communication subspace” to “cross-area subspace” or “CCA subspace” throughout the manuscript.

- Line 882: References 10-12 are cited for CCA. This might be a typo.

We thank the reviewer for catching this error, the citations have been updated accordingly.

- Line 885: "mean activity in each group was subtracted." The authors should justify why this is needed and how it affects interpretation.

Subtracting mean firing rate is a defined step in the canonical correlation analysis, as the process involves comparing data with zero mean and unit standard deviation; it is built into the MATLAB function *canoncorr* (Guo, et. al., Cell Reports 2021, Veuthy* & Derosier, et. al, Nature Communications 2020). Theoretically, by mean subtracting, one can identify patterns of communication to downstream regions that are independent of absolute firing rate. We have clarified this point in Methods, Cross-area neural subspace (**lines 939-941**).

- Supp Fig 3B: Explain what the reader is supposed to take away from these CCA R^2 values. Also, it's unclear how they support the title of this figure: "Majority of M1 and DLS spiking not related to reach direction."

The R^2 values measure the predictive power of the model, that is how well the model generalizes to held out data. These R^2 values come from cross-validation – we randomly partition the full dataset into 10 folds and cycle through each fold, assigning one fold to be the test data and the other nine to be the training data. We fit a CCA model to the training data, then project the test data onto this model, and compute the R^2 between the M1 and DLS projections. We find that R^2 is consistent across sessions, leading us to conclude that communication along the M1 and DLS CCA subspace occurs at the same “strength” across the relearning

paradigm. We have clarified these points in Methods, Cross-area neural subspace (**lines 942-949**) and Results, Changes in M1-DLS cross-area subspace modulation (**lines 401-405**).

We thank the reviewer for pointing out that the figure title does not accurately reflect the data communicated by CCA R^2 values. We have moved this panel to Supplemental Figure 7a (**lines 1320-1322**).

REVIEWER COMMENTS

Reviewer #1 (Remarks to the Author):

The authors have made substantial changes to this paper in responses to the referees' comments. I have no detailed comments.

Reviewer #2 (Remarks to the Author):

The authors have responded well to the criticisms and improved the manuscript. The added details of the behavioral experiments make the work easier to understand and the handling of the analyses are better described in this new version.

Reviewer #3 (Remarks to the Author):

My comments/questions have been adequately addressed. I really enjoyed this paper and think it adds substantially to the literature on understanding the control of coordinated skills.

Reviewer #4 (Remarks to the Author):

The authors have done a good job with the revisions. A few remaining clarifications/comments:

1) Supp Fig 7a: More could be done to guide the reader through this figure, both regarding basic details and -- more importantly -- the primary conclusion of the analysis represented.

Regarding basic details: What do individual points represent? Why are there a different number of points in each panel (one can infer the answer after reading L948-954 of Methods, but descriptions here would be helpful).

Regarding the primary conclusion: If I did not read the authors' conclusion, the most salient feature to me is the variability in R^2 values. Take, as one example, the "rel" column of models fit to B-reach trials: predictive performance varies by a factor of 5 between the least predictive and most predictive models. Could the authors clarify why one should conclude from these plots that the M1-DLS CCA subspace is "similarly generalizable across sessions"?

2) Could the authors more carefully define "peak subspace modulation" (used, e.g., in L423-424, L963, Supp Fig 7bc)? How does this metric differ from the original RMI metric (Fig 5e), if at all?

3) The RMI metric should be explained in more detail in Methods. In the authors' response, they write: "In those trials where the RMI is negative, the projected trial spiking on CV1 is less correlated during reach than it is prior to reach." To clarify, it is not that the activity is less correlated, but rather the activity pre-reach is higher than the activity during reach, correct? If this is true, it would be helpful to make this clear to the reader.

Minor comments:

- L402: "whether the CCA model generalizability across days is comparable". This sounds like a CCA model is being fit to day 1, and then applied to day 2. Clarify what this means.

- L435: "there was not significant..." should read "there was less significant..."

- L512: Given the authors' change in terminology throughout the manuscript, "communication space" should be changed to "CCA subspace" (or similar) for consistency.

- L1218 (Fig 5 legend): "maximal correlation" would be a more appropriate term than "maximal shared variance".

INTRODUCTION TO REVISED MANUSCRIPT

We thank the reviewers (R1/R2/R3/R4) for their thorough review and comments, which have significantly strengthened our approach. We also greatly appreciate the general enthusiasm for our study! We outline specific changes in response to the comments below. We also appreciate the consideration of the manuscript by the editors and have included a response regarding the submission at the end.

Detailed response to Reviewer #1:

“The authors have made substantial changes to this paper in responses to the referees' comments. I have no detailed comments.”

We thank for reviewer for their positive comments.

Detailed response to Reviewer #2:

“The authors have responded well to the criticisms and improved the manuscript. The added details of the behavioral experiments make the work easier to understand and the handling of the analyses are better described in this new version.”

We thank for reviewer for their positive comments and appreciate that the revised analysis descriptions have improved the manuscript.

Detailed response to Reviewer #3:

“My comments/questions have been adequately addressed. I really enjoyed this paper and think it adds substantially to the literature on understanding the control of coordinated skills.”

We thank for reviewer for their positive comments and appreciate the feedback regarding how this study adds to the coordinated skill control literature.

Detailed response to Reviewer #4:

"The authors have done a good job with the revisions. A few remaining clarifications/comments:"

We thank for reviewer for their positive comments and appreciate the feedback regarding how to make the descriptions clearer.

"1) Supp Fig 7a: More could be done to guide the reader through this figure, both regarding basic details and -- more importantly -- the primary conclusion of the analysis represented."

"Regarding basic details: What do individual points represent? Why are there a different number of points in each panel (one can infer the answer after reading L948-954 of Methods, but descriptions here would be helpful)."

There are a different number of points in each panel, because only sessions with enough units (at least 5 in each region) to run CCA analyses and models with significant R^2 values were included. These were built separately for reaches to A versus B – therefore if a relearned session no longer had reaches to A, a CCA model for reaches to A during the relearned session type could not be built. We have updated the figure legend (**Lines 1324-1325**) to clarify that the individual points represent top CV R^2 values only from sessions that met the CCA inclusion criteria as detailed here and in the Methods.

"Regarding the primary conclusion: If I did not read the authors' conclusion, the most salient feature to me is the variability in R^2 values. Take, as one example, the "rel" column of models fit to B-reach trials: predictive performance varies by a factor of 5 between the least predictive and most predictive models. Could the authors clarify why one should conclude from these plots that the M1-DLS CCA subspace is "similarly generalizable across sessions"?"

We thank the reviewer for bringing up the alternate interpretation of the plots. R^2 values can range from 0 to 1, and the scales of the graph may in part have created the appearance of strong variability. Because we have only focused on the automatic, variable, and relearned sessions for CCA analyses, we have modified Supplemental Figure 7a to be more representative of the data: in the left panel, we demonstrate R^2 values for CCA models of reach to A across the automatic, variable, and relearned session types, with the right panel showing the variable and relearned sessions for reaches to B. Additionally, we have added the mean and standard deviation to better enable comparison of R^2 across session types. There is no significant difference in R^2 values across different session types demonstrated, for models built to A or models built to B. Thus, the reliability of the CCA model in being generally applicable is not significantly different for a given session type.

Of note, the number and quality of units in each region, that can vary from day to day, can contribute to variability of the R^2 value. The alternative hypothesis would be that as we see breakdown in spatiotemporal spiking consistency in both M1 and DLS during the variable session, CCA models could have a lower R^2 during this day as well, if M1-DLS correlated activity was not maintained in some structured fashion. However, we do not see evidence of this in the data we have. That being said, we have modified the Supplemental Figure 7 description to more broadly capture that the panels represent further CCA analyses beyond Figure 5.

"2) Could the authors more carefully define "peak subspace modulation" (used, e.g., in L423-424, L963, Supp Fig 7bc)? How does this metric differ from the original RMI metric (Fig 5e), if at all?"

We thank the reviewer for raising this point. Peak subspace modulation is defined as the maximum subspace activity during -0.2s to 0.2s around reach onset, whereas the relative modulation index is the difference in median subspace activity from the reach period (-0.1s to 0.4s around reach onset) compared to the pre-reach period (-1s to -0.5s before reach onset). We have clarified the definition of peak subspace modulation in the main text (**Lines 424-425**), Methods (**Lines 974-976**), as well as Supplemental Figure 7 legend (**Lines 1327-1328**).

"3) The RMI metric should be explained in more detail in Methods. In the authors' response, they write: "In those trials where the RMI is negative, the projected trial spiking on CV1 is less correlated during reach than it is prior to reach." To clarify, it is not that the activity is less correlated, but rather the activity pre-reach is higher than the activity during reach, correct? If this is true, it would be helpful to make this clear to the reader."

We thank the reviewer for pointing out that the RMI metric was unclearly defined in the Methods and have clarified this (**Lines 969-970**). For the second comment, we thank the reviewer for pointing out the nuance

regarding “correlation” versus “CCA subspace activity” and agree that with a negative RMI, there is greater CCA subspace activity during the pre-reach period as compared to the reach period. We have updated the Methods to include interpretation of the RMI metric (**Lines 973-974**).

Minor comments:

“- L402: "whether the CCA model generalizability across days is comparable". This sounds like a CCA model is being fit to day 1, and then applied to day 2. Clarify what this means.”

The CCA model is fit separately for each session, as the units cannot be verified as being identical across sessions. The clarification regarding CCA model generalizability is stated after the semicolon – “that is, does CCA model R2, a measure of the predictive power of the model, remain similar across days?” (**Lines 403-404**).

“- L435: "there was not significant..." should read "there was less significant..."”

We thank the reviewer for the edit and have updated the sentence accordingly (**Line 436**).

“- L512: Given the authors' change in terminology throughout the manuscript, "communication space" should be changed to "CCA subspace" (or similar) for consistency.”

We appreciate the reviewer pointing this out and have updated the sentence for consistency (**Line 514**).

“- L1218 (Fig 5 legend): "maximal correlation" would be a more appropriate term than "maximal shared variance".”

We appreciate the clarification and have updated the terminology accordingly (**Line 1233**).